# Soil organic carbon mineralization is controlled by the application dose of exogenous organic matter

Orly Mendoza [1], Stefaan De Neve [1], Heleen Deroo [1], Haichao Li [1], Astrid Françoys [1,2], Steven Sleutel [1]

[1] Department of Environment, Ghent University, Coupure Links 653, 9000 Ghent, Belgium
[2] Isotope Bioscience Laboratory, Department of Green Chemistry and Technology, Ghent University, Coupure Links 653, 9000 Ghent, Belgium

*Correspondence to:* Orly Mendoza (mendozaorly@gmail.com)

**Abstract.** Substantial input of exogenous organic matter (EOM) may be required to offset the projected decline in soil organic carbon (SOC) stocks in croplands caused by global warming. However, information on the effectivity of EOM application dose in preserving SOC stocks is surprisingly limited. Therefore, we set up a 90-day incubation experiment with large soil volumes (sandy loam and silt loam) to compare the mineralization of EOM ($^{13}$C-labelled ryegrass) and SOC as a function of three EOM application doses (0.5, 1.5, and 5 g dry matter kg$^{-1}$ soil). In the sandy loam soil, the percentage of mineralized EOM was not affected by EOM dose, while SOC mineralization increased proportionally with increasing EOM dose (+49.6 mg C per g EOM). Likewise, formation of MBC was proportional to EOM dose, suggesting no increased microbial economic growth at higher C concentration. In the silt loam soil, the percentage of mineralized EOM decreased somewhat with increasing dose, while SOC mineralization increased at a higher rate than in the sandy loam soil (+117.2 mg C per g EOM). Possibly the higher microbial activity might have lowered soil $O_2$ supply close to the large added EOM particles, limiting its relative degradability, although this was not suggested from bulk soil Eh data. In both textured soils, increasing EOM dose possibly supplied energy for microbial growth and enzyme production, which in turn stimulated mineralization of native SOC (i.e. co-metabolism). The observed stimulation of soil macroporosity at higher EOM doses in the silt loam soil could have contributed to sustaining aerobic conditions required for SOC mineralization in the silt loam soil. At the same time the higher microbial activity might have sufficiently lowered soil Eh close to the large added EOM particles, limiting its relative degradability at high dose, suggesting a potential new mechanism for understanding SOC cycling. In sum, this experiment and our previous research suggest that EOM mineralization is mostly independent of EOM dose, but EOM dose modulates mineralization of native SOC. Provisional C balances compared to unamended controls indicated that at low doses, less C remained than when EOM was added at normal or high doses. These findings tentatively indicate that using larger EOM doses could help preserve more of added EOM-C, but longer-term confirmation in the field will firstly be required before we could draw any conclusion for soil C management.

## 1    Introduction

Small changes in global carbon (C) stocks can cause significant changes in climate (Smith et al., 2020). Croplands are a potential global C sink because of their lower soil organic carbon (SOC) content relative to that of the corresponding native ecosystems (Paustian et al., 1997). Zomer et al. (2017) estimated that croplands could potentially sequester 0.90–1.85 Gt C year$^{-1}$, representing a substantial portion (i.e. 26–53 %) of the 4p1000 initiative target (3.4 Gt C year$^{-1}$) that aims at offsetting most of the annual increase in atmospheric $CO_2$ (15.8 Gt $CO_2$ year$^{-1}$ or 4.3 Gt C year$^{-1}$) (Paustian et al., 2019). To preserve SOC stocks and soil fertility, most agricultural systems rely on the application of exogenous organic matter (EOM) to the soil, usually as crop residue or animal manure. Depending on its composition and dose, EOM contributes significantly to the overall soil C balance, which can be derived using soil C balance calculations or simulation model runs. In both empirical and more complex process-based SOC simulation models, EOM degradability is primarily

determined by its quality, soil texture, and soil environmental parameters, such as temperature, moisture content, and availability of N (Kutsch, 2012). However, the EOM dose (amount of C added per kg soil or per m$^2$) has supposedly little or no impact on its mineralization rate or on the mineralization rate of SOC in SOC models. For example, the Roth-C model (Jenkinson et al., 2008; Powlson et al., 2013) and the AMG model (Andriulo et al., 1999) simulate an unrestricted response of C mineralization to C concentration. The DNDC model simulates several anaerobic processes via Michaelis-Menten kinetics (Li et al., 1997), i.e. with a feedback to exogenous organic carbon (EOC) dosage, but again the aerobic mineralization of C follows first-order kinetics.

Nevertheless, several reasons have been provided for feedback between the EOM dose and its decomposition in soil. First, the fate of OM in soil can be controlled by its accessibility to decomposers (Dungait et al., 2012; Lehmann et al., 2020). Thus, larger soil EOM quantities may promote closer contact with decomposers and positive impacts on EOM and SOC mineralization. Don et al. (2013) demonstrated that total SOC was more readily mineralized when the added EOM (compost) was concentrated as opposed to when it was dispersed in soil. Kuzyakov and Domanski (2000) also argued that the local disproportional growth of microbial biomass and stimulation of its activity with EOM dose may have positive effects on EOM and SOC decomposition via co-metabolism or other SOC priming mechanisms. However, negative SOC priming might also occur if, for instance, higher stimulated heterotrophic activity leads to local depletion of O$_2$ during decomposition, slowing down further EOM mineralization. Second, stimulated microbial activity at higher EOM doses might indirectly modulate soil microbial activity by controlling soil structure development. De Gryze et al. (2005) reported that increasing the doses of wheat residue led to a proportional formation of soil aggregates. Furthermore, Shahbaz et al. (2017a) concluded that adding a higher dose of wheat residue (1.40 vs. 5.04 g dry matter kg$^{-1}$) stimulated macroaggregate formation, resulting in positive priming of SOC in a silt loam soil. Using the same soil but higher wheat residue doses (5.4 and 10.8 g kg$^{-1}$ soil vs. 1.40 and 5.04 g kg$^{-1}$ soil), Shahbaz et al. (2017b) also reported that SOC priming diminished at high dose. Similarly, Mendoza et al. (2022a) observed that increasing EOM (ground maize straw and ryegrass) doses stimulated soil macroporosity but did not affect EOM mineralization. Such control of soil structure and potential feedback to EOM and SOC mineralization can be soil specific. Increasing the application dose of $^{13}$C-labelled ryegrass promoted the formation of meso- and macropores, and their volume percentages correlated positively with the magnitude of positive SOC priming effects in a sandy loam but not in a silt loam soil (Mendoza et al., 2022b). Finally, adding relatively N-poor EOM at a higher dosage can adversely impact its degradation due to the temporal shortage of mineral N in soil for microbial decomposers.

The conclusions of the above-mentioned studies, focusing on C mineralization in response to the application dose, were mostly based on simplified soil systems. Despite the diverse chemical complexity encountered in real conditions, laboratory experiments studying dosage effects on C mineralization are often limited by the use of single chemical compounds, such as glucose (Blagodatskaya et al., 2007; Liu et al., 2017; Schneckenberger et al., 2008). More complex plant-derived substrates were finely ground and mixed in soil, such as <2 mm wheat residue (Shahbaz et al., 2017a, b), ground maize straw particles (Mendoza et al., 2022a), and ±2 cm chopped pieces of ryegrass (Mendoza et al., 2022b). Such finer EOM sources likely decompose differently in laboratory than in field conditions because of the large surface area of the residue in contact with the soil (Garnier et al., 2008); on the other hand, a potentially stronger interaction with the soil mineral phase may protect OM from decomposition. Both phenomena, using single chemical compounds and their unrepresentative small size compared to more complex and larger-sized plant residue pieces at field conditions, may render the EOM dosage responses from laboratory incubations unrepresentative of OM degradation in the field. To better approach field conditions incubation experiments would need to be upscaled with larger pieces of crop residue that could constitute C hotspots with a more locally confined but stronger impact on soil structure and biological soil processes.

Moreover, at such C hotspots, oxygen shortages may impede aerobic soil C mineralization. Such responses have been largely overlooked when interpreting soil C mineralization data (Keiluweit et al., 2017), and soil structure should thus also be kept as intact as possible to account for those aspects.

To account for these shortcomings and to obtain more realistic understanding of the application dose effect on EOM and SOC mineralization as it might occur in the field, we aimed to study decomposition of large pieces of $^{13}$C-labelled ryegrass residue in large relatively less disturbed soil cores. Consequently, as compared to Mendoza et al. (2022b) and Shahbaz (2017a), the soil masses used in a newly designed soil incubation experiment were about 23 times and 70 times larger, the soils were only coarsely sieved (<10 mm), the added crop residues were not chopped. Concentration of added ryegrass 90 dose to non-refined ryegrass pieces is expected to yield a stronger local impact on soil structure and the concentration of C mineralization into such relative hotspots might more readily result in local depletion of $O_2$ with potential negative impacts on EOM and SOC mineralization as compared to previous work on this topic. We hypothesized that the mineralized percentage of added EOM (further referred to as relative EOM mineralization) would increase with increasing application dose. To gain insight into the potential indirect control of EOM dose on its mineralization via the mediation 95 of soil aeration and structure, the effects of EOM dosage on soil redox potential and pore neck size distribution were assessed. The experiment was performed in both sandy loam and silt loam soils. We expected that in finer-textured soil, $O_2$ provision would more readily limit EOM mineralization at higher EOM doses because of slower gaseous diffusion compared to in coarser-textured soil. We also hypothesized that SOC mineralization would be stimulated as a larger EOM application dose would lead to SOC priming, which can be linked to the classical co-metabolism mechanism and to 100 promoted soil aeration likely by enhancement of macroporosity. An overview of the hypotheses can be seen in Fig.1.

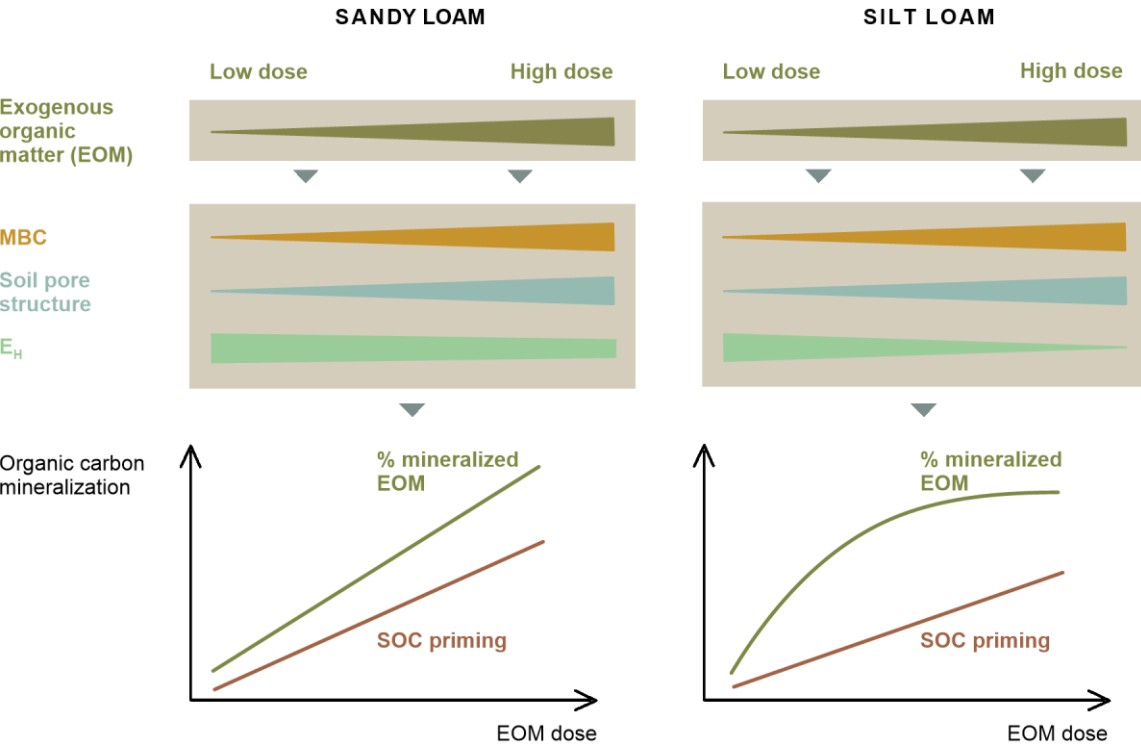

**Figure 1.** Expected outcomes and hypotheses of the EOM application dose effect on EOM and native SOC mineralization in a sandy loam and a silt loam soil.

## 2   Materials and methods

**2.1  Soils and labelled $^{13}$C ryegrass used in the incubation experiments**

A soil incubation experiment was set up with different EOM doses and soil textures to investigate their interactive effect on EOM-C mineralization. Two topsoils were selected with contrasting soil texture, i.e. sandy loam and silt loam, but with similar SOC content, C:N ratio (10 and 9, respectively), and $\delta^{13}$C (Table 1). These soils were sampled at a depth of approximately 5 to 20 cm at the Institute for Agricultural Research in Melle, Belgium and at a nearby farmer's field in Oosterzele, East Flanders, Belgium.

$^{13}$C-pulse-labelled ryegrass was used as the model EOM for a reliable discrimination of EOM and SOC mineralization based on $^{13/12}$C-resolved soil $CO_2$ efflux measurements. $^{13}$C-pulse-labelling was performed at the Ghent University experimental farm in Bottelare, Belgium by weekly exposure of initially pruned ryegrass (*Lolium perenne* L.) for two months to a temporarily $^{13}CO_2$-enriched atmosphere inside Plexiglas labelling chambers. These $^{13}CO_2$ flushes were obtained by reacting 0.3 M sodium bicarbonate-$^{13}$C (NaH$^{13}$CO$_3$ 98 atom % $^{13}$C) with 1 M hydrochloric acid (HCl) inside the chambers that were outfitted with fans, sealed on the ground, and closed overnight on top of 0.8×0.8 m microplots in the ryegrass field. After 8 weeks, ryegrass was cut with a sickle and had a resulting $\delta^{13}$C of +51±5.4 ‰ (n = 6), C content of 439.3±5.4 g kg$^{-1}$, and C:N of 10±0.5. Harvested and dried $^{13}$C-labelled ryegrass from young and more senescent plant parts differed by only 3.8‰, demonstrating that the material was uniformly $^{13}$C labelled.

**Table 1.** Physicochemical characteristics, and current and past crops of the two selected topsoils used in the soil incubation experiment. Averages of three replicates with standard errors are shown for pH, and measurements of two replicates are shown for Al$_{ox}$, Fe$_{ox}$, P, Ca, K, and Mg.

| Soil characteristic | Sandy loam | Silt loam |
|---|---|---|
| Sand (%) | 60.2 | 13.3 |
| Silt (%) | 26.9 | 67.0 |
| Clay (%) | 12.9 | 19.7 |
| SOC (g kg$^{-1}$) | 14.4±0.4 | 14.9±0.3 |
| $\delta^{13}$C (‰) | −27.72 | −25.11 |
| pH$_{(H2O)}$[a] | 6.4±0.0 | 7.4±0.0 |
| Al$_{ox}$ (mg kg$^{-1}$)[b] | 1199.4 – 1172.7 | 679.7 – 693.7 |
| Fe$_{ox}$ (mg kg$^{-1}$)[b] | 3221.0 – 3183.0 | 4280.5 – 4302.0 |
| P (mg kg$^{-1}$)[c] | 85.7 – 87.8 | 325.6 – 329.0 |
| Ca (mg kg$^{-1}$)[c] | 954.5 – 953.8 | 2429.2 – 2488.0 |
| K (mg kg$^{-1}$)[c] | 148.0 – 147.5 | 320.5 – 327.2 |
| Mg (mg kg$^{-1}$)[c] | 125.2 – 125.0 | 142.4 – 146.3 |
| Soil type | Cambisol | Cambisol |
| Current crop | French marigolds (*Tagetes patula* L.) | Maize (*Zea mays* L.) |
| Cropping history | 2012-2019: ryegrass | 2012-2019: potato, wheat, barley, maize, sugar beets, potato, wheat |

[a]pH$_{(H2O)}$ was obtained by inserting a glass pH electrode in 1:6.25 soil-H$_2$O mixture. [b]NH$_4$-oxalate extraction and detection using inductively coupled plasma optical emission spectroscopy (ICP-OES). [c]NH$_4$–acetate-EDTA extraction and ICP-OES analysis.

## 2.2 Soil incubation and experimental design

The soils collected from the field were kept moistened and sieved through a 10 mm mesh to keep initial soil aggregation as intact as possible. Moist soil directly limited the disruption of the original soil macro-aggregation during sieving. The collected air-dried ryegrass biomass (no further refinement after harvesting) was mixed into 7.96 kg of moist sandy loam soil and 7.78 kg of moist silt loam soil at application doses of 0.5, 1.5 and 5 g dry matter $kg^{-1}$ soil and transferred into large PVC pots (Ø 19.1 cm and height 25 cm). These applications were used as representative for rather low, intermediate, and high EOM doses commonly applied in agricultural practices under field conditions, and they correspond to 1.75, 5.25, and 17.5 Mg $ha^{-1}$ assuming a depth of 25 cm and a bulk density of 1.4 g $cm^{-3}$. The intermediate dose of 1.5 g dry matter $kg^{-1}$ soil furthermore closely represents the typical application rate of 2.6 Mg C $ha^{-1}$ $yr^{-1}$ in German croplands (Riggers et al., 2021). The experiment was limited to only three doses because the large dimensions of the soil mesocosms (6.5-7 kg dry soil per pot) already required considerable amounts of $^{13}$C-labelled plant material. Unamended controls were also included for both soil textures. Based on pre-tests, soil bulk densities of 1.35 and 1.25 g $cm^{-3}$ were employed for the sandy loam and silt loam soils, respectively, as soil structural integrity was sufficiently intact at these densities, while minimal soil compaction was required for core filling. The sandy loam and silt loam soils were amended with 34 and 20 mg $NO_3^-$-N $kg^{-1}$, respectively, by mixing a $KNO_3$ solution to adjust the soil N concentration to 35.7 mg $NO_3^-$-N $kg^{-1}$ (equivalent to 150 kg N $ha^{-1}$ as per surface area assuming a depth of 30 cm and a bulk density of 1.4 g $cm^{-3}$). By adding mineral N, we aimed to exclude the possibility of dosage effects by differences in soil mineral N availability. Soil moisture was maintained at 55% water-filled pore space by adding demineralized water to the top of the soil cores. The packed soil mesocosms were covered with perforated parafilm to enable gas exchange but limit evaporation and were stored in a dark room at a constant temperature of 20±1.0 °C. Soil moisture was kept constant throughout the incubation experiment by regularly weighing the soil pots and replenishing evaporation water loss with demineralized water. The experiment comprised two soil textures × (three EOM application doses + unamended controls) × three replicates, yielding 24 soil mesocosms.

### 2.3 Soil $CO_2$ efflux measurements, isotopic analysis and source partitioning of soil C mineralization

Soil $CO_2$ efflux rates were obtained 14 times during the 90-day experiment by measuring the $CO_2$ build-up over time in an opaque closed PVC chamber headspace (8.45 l) on top of each of the PVC tubes with the soil mesocosms. The closed chamber was outfitted with a battery-powered fan and a pressure vent. Changes in headspace $CO_2$ concentrations and their $\delta^{13}$C (in ‰ relative to the international Vienna Pee Dee Belemnite standard) resulting from soil $CO_2$ emissions were measured in real time by consecutively connecting a cavity ring-down spectroscopy analyser (G2201-i CRDS isotopic $CO_2$/$CH_4$ analyser, Picarro, United States) to each headspace chamber. A linear increase in $CO_2$ was recorded every 1–2 s for ~10 min or less to avoid excessive build-up of headspace $CO_2$ which could cause a drop in the diffusion gradient within the soil in treatment/time combinations during high microbial respiration. A linear model was fitted to the observed increase in headspace $CO_2$ concentration, and the soil $CO_2$ efflux rate was obtained from its slope and converted into a mass-based unit (mg $kg^{-1}$ $day^{-1}$) using the ideal gas law. $\delta^{13}$C of the emitted $CO_2$ was determined as the intercept of the linear regression line between the headspace air $\delta^{13}$C and reciprocal of the headspace $CO_2$ concentration (Keeling, 1958).

The efflux of $CO_2$ derived from ryegrass mineralization was calculated by the following isotopic mixing model (Mendoza et al., 2022a):

$$CO_2\text{-}C_{(ryegrass)} = CO_2\text{-}C * \frac{\delta^{13}C\text{-}CO_2 - \delta^{13}C\text{-}CO_{2(0)}}{\delta^{13}C\text{-}CO_{2(ryegrass)} - \delta^{13}C\text{-}CO_{2(0)}} \tag{1}$$

where $CO_2$-$C$ is the overall soil $CO_2$-C efflux rate, $\delta^{13}C$-$CO_2$ is the isotopic signature of the respired $CO_2$ estimated using the Keeling plot, $\delta^{13}C$-$CO_{2(ryegrass)}$ is the estimated isotopic signature of emitted $CO_2$ resulting from ryegrass

mineralization (Eq. 2), and $\delta^{13}C\text{-}CO_{2(0)}$ is the isotopic signature of $CO_2$ measured from treatments with no ryegrass added (i.e. resulting from SOC mineralization only).

Isotopic fractionation of $CO_2$ caused by either microbial mineralization or diffusive transport to the headspace air was estimated separately for SOC and ryegrass EOM. The isotopic signature of the SOC-derived $CO_2$ measured in the unamended treatments was used to estimate C isotopic fractionation during SOC mineralization and diffusive SOC-derived $CO_2$ transport to the headspace air. The isotopic fractionation of ryegrass-C mineralization and its diffusive transport to the headspace air was estimated as the shift in $\delta^{13}C$ of the $CO_2$ emitted from the highest EOM dose (i.e. from ryegrass + SOC) and $CO_2$ emissions from the unamended control (i.e. from SOC), following a mass balance analogous to Keeling (1958), as follows:

$$\delta^{13}C\text{-}CO_{2(ryegrass)} = \frac{CO_2\text{-}C_{(5)} * \delta^{13}C\text{-}CO_{2(5)} - CO_2\text{-}C_{(0)} * \delta^{13}C\text{-}CO_{2(0)}}{CO_2\text{-}C_{(5)} - CO_2\text{-}C_{(0)}} \tag{2}$$

where $CO_2\text{-}C_{(5)}$ and $CO_2\text{-}C_{(0)}$ are the total $CO_2$-C fluxes corresponding to ryegrass doses of 5 and 0 g kg$^{-1}$ soil measured at each soil texture, and $\delta^{13}C\text{-}CO_{2(5)}$ and $\delta^{13}C\text{-}CO_{2(0)}$ are the respective $\delta^{13}C\text{-}CO_2$.

The $CO_2$-C derived from SOC mineralization in the amended soils, $CO_2\text{-}C_{(SOC)}$ (mg $CO_2$-C kg soil day$^{-1}$), was calculated as follows:

$$CO_2\text{-}C_{(SOC)} = CO_2\text{-}C - CO_2\text{-}C_{(ryegrass)} \tag{3}$$

Finally, the cumulative amounts of mineralized ryegrass C and SOC were calculated. The relative priming effect (PE) of SOC mineralization induced by EOM application dose was calculated for each soil texture as follows:

$$PE_{(SOC)} = \frac{CO_2\text{-}C_{(soc)} - CO_2\text{-}C_{(0)}}{CO_2\text{-}C_{(0)}} * 100 \tag{4}$$

## 2.4 Microbial biomass carbon and soil mineral nitrogen

Eight additional soil mesocosms (four EOM doses: 0, 0.5, 1.5, and 5 g kg$^{-1}$ soil × two soil textures: sandy loam and silt loam) were prepared as described in Section 2.2, but employing non-labelled ryegrass grown under the same environmental and edaphic conditions as the $^{13}$C-labelled grass. After 45 days, soil was sampled, and microbial biomass carbon (MBC) was quantified using the fumigation-extraction method (Vance et al., 1987). We measured MBC after 45 days because then we observed that most of the SOC priming and EOM mineralization had occurred. Before and after fumigating fresh soil samples with ethanol-free chloroform, 30 g of soil was extracted with 60 ml 0.5 M $K_2SO_4$. These extracts were passed through a filter (Whatman 5) and analysed for their dissolved OC concentration using a TOC analyser (TOC/TN analyser, Skalar, The Netherlands). MBC was determined as the difference in extractable C between fumigated and non-fumigated soils, and corrected with a factor of 0.45 to account for the MBC fraction extractable by fumigation (Joergensen, 1996). The EOM-mediated MBC increase (i.e. difference in MBC between amended and control divided by application dose) was also calculated for each application dose to test whether MBC was proportionally stimulated per unit of EOM added.

At the end of the 90-day incubation experiment, soil from each mesocosm was destructively sampled and mineral N was determined. The soil was taken out of the PVC pots, homogenised, from which 20 g was weighed and shaken with 100 ml 1 M KCl for one hour, and the extracts were filtered (Macherey-Nagel, USA, MN 616 1/4, Ø 150 mm filters). $NH_4^+$-N and $NO_3^-$-N concentrations in the extracts were measured using a continuous-flow analyser (Skalar, San++ Continuous Flow Analyser, Netherlands). Soil $NH_4^+$-N was negligible in all treatments; therefore, only $NO_3^-$-N concentrations were considered.

## 2.5 Soil redox potential

The soil redox potential (Eh) was measured 21 times during the 90-day incubation experiment in each of the 24 mesocosms. Every soil core was permanently outfitted with one Ag|AgCl saturated KCl reference electrode and a redox probe (Paleoterra, Netherlands). The redox probes consisted of two platinum sensors (surface area ±5 mm² each) situated 5 cm and 15 cm below the soil surface. The reference electrodes and redox probes were first tested and calibrated with 124 and 250 mV redox standards (Sigma-Aldrich, Ireland). Soil Eh readings were obtained by connecting the reference and Pt electrodes through a high-impedance redox mV meter (Paleoterra, The Netherlands). The Eh readings were expressed versus the standard hydrogen electrode after temperature and offset corrections with the Ag|AgCl reference electrode by adding 204 mV to the mV readings.

## 2.6 Pore neck size distribution

After 45 days, four intact soil cores were carefully sampled using stainless-steel sampling rings (Ø 5.0 cm and height 5.1 cm) from each of the eight mesocosms described in Section 2.4. These rings were used to determine the soil water retention curve using the sandbox pressure plate method. After fitting a nylon mesh onto the bottom of the rings, they were placed on a sand box apparatus (Eijkelkamp Agrisearch Equipment, Giesbeek, Netherlands), saturated, subjected to pressure potentials of -10, -30, -50, -70, and -100 hPa, and weighed for the corresponding soil water content after reaching equilibrium. The soil water content at soil matric potentials of -330, -1000 and -15000 hPa were determined using pressure plates (Eijkelkamp Agrisearch Equipment, Giesbeek, Netherlands and Soil Moisture Equipment, Santa Barbara, CA, USA). Volume of the soil pore neck size classes corresponding to these water potentials was calculated using Jurin's law: d = -3000/$\Psi$, where $d$ is the pore neck diameter (µm) and $\Psi$ the matric potential (hPa) (Schjønning et al., 1999). Consequently, the volume proportion (vol%) of pores with pore neck diameters of >300, 100–300, 60–100, 43–60, 30–43, 9–30, 3–9, 0.2–3, and <0.2 µm were calculated.

## 2.7 Statistical analyses

General linear models (GLMs) were used to evaluate the EOM dose-response relationships at a single time point, whereas linear mixed models (LMMs) were used to assess the EOM dose effects over time. Specifically, GLM were employed to assess the effects of EOM dose and soil texture on EOM mineralization, SOC mineralization and mineral N at the end of the incubation, and on MBC and EOM-mediated MBC increase as a function of dose at day 45 of the incubation. If the GLM showed a significant interaction between EOM dose and soil texture, separate GLMs with different EOM doses were conducted per soil texture. In the case of no interaction, the interaction term was removed from the model and an additive model was built with the EOM dose and soil texture. Additionally, one-way analysis of variance (ANOVA) was conducted for each soil texture to check the effects of EOM dose on the vol% of pore neck size classes. On the other hand, LMMs were used to compare the effects of EOM dose on soil Eh over time with the fixed effects of EOM dose, measurement depth, interaction depth × dose, and random effects of time and replicate. Normality was assessed by the QQ plots of the residuals, and homoscedasticity was verified by plotting residuals versus fitted values. The Shapiro-Wilk's test and Levene's test were used to confirm the normality of the distributed model residuals and equality of variances, respectively. When those assumptions were not met, log transformations were applied to our data. Pearson's correlation coefficients were calculated for the relative SOC priming and MBC. All statistical tests were conducted with R version 3.6.1, using the packages car and agricolae for GLM and one-way ANOVA, lme4 for LMM, and hmisc and ggpubr to detect correlations between the variables.

## 3 Results

### 3.1 EOM mineralization

The amount of mineralized EOM increased linearly after 90 days with the EOM application dose for both textures ($P$ <0.001), with determination coefficients close to 1. Less EOM was mineralized in the sandy loam than in the silt loam soil, with values of 223 and 279 mg C per added gram of EOM, respectively (Fig. 2 top). The relative fraction of added EOM mineralized after 90 days depended on the soil texture. In the sandy loam soil, it was independent of EOM application dose, whereas in the silt loam soil, the relative fraction of mineralized EOM decreased with increasing EOM dose ($P$ <0.05; Fig. 2 bottom).

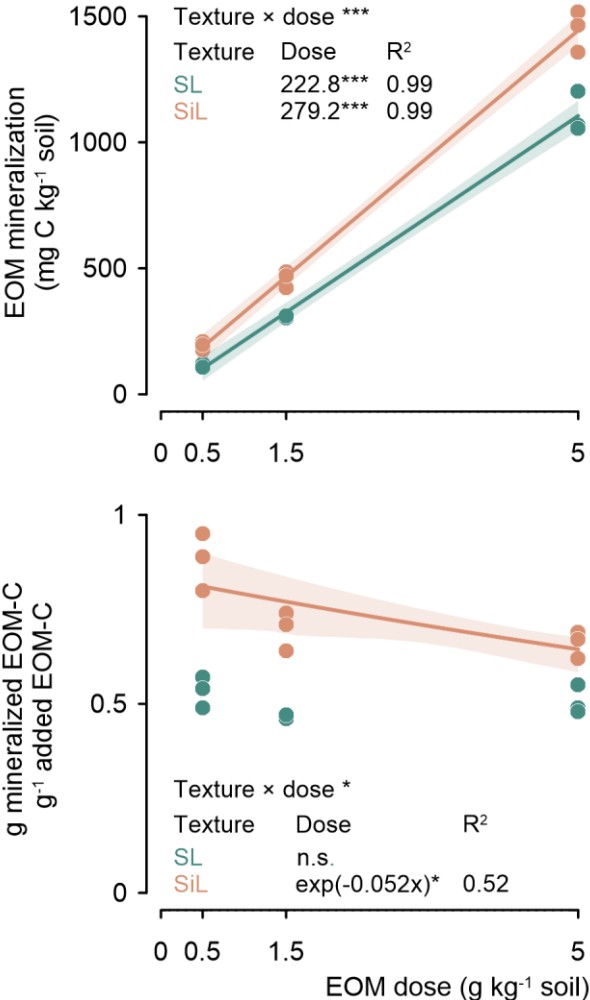

Figure 2. Cumulative mineralized EOM C (ryegrass) after 90 days since incorporation in sandy loam (SL) or silt loam (SiL) soil as a function of its application dose (upper figure). The lower figure presents the proportion of mineralized ryegrass C as a function of EOM dose. * denote significance levels of the linear model interaction effect between soil texture and EOM dose, and for the linear or exponential response of EOM mineralization to EOM dose, with *** <0.001, ** <0.01, * <0.05, and n.s. = not significant. Polygons around regression lines represent 95% confidence intervals.

### 3.2 SOC mineralization

After 90 days of incubation, the cumulative mineralized SOC increased linearly with increasing application dose in both soils and this dosage response was stronger in the silt loam soil (117.2 mg C per g EOM) ($P$ <0.001) compared with the sandy loam soil (49.6 mg C per g EOM) (Fig. 3 top).

EOM application dose also stimulated the relative priming of native SOC mineralization (i.e. difference of SOC mineralization between amended and unamended control relative to the unamended control). Such positive relative SOC

priming increased by 12.6 % per g EOM added in both the sandy loam and silt loam soil textures ($P$ <0.001; Fig. 3 bottom).

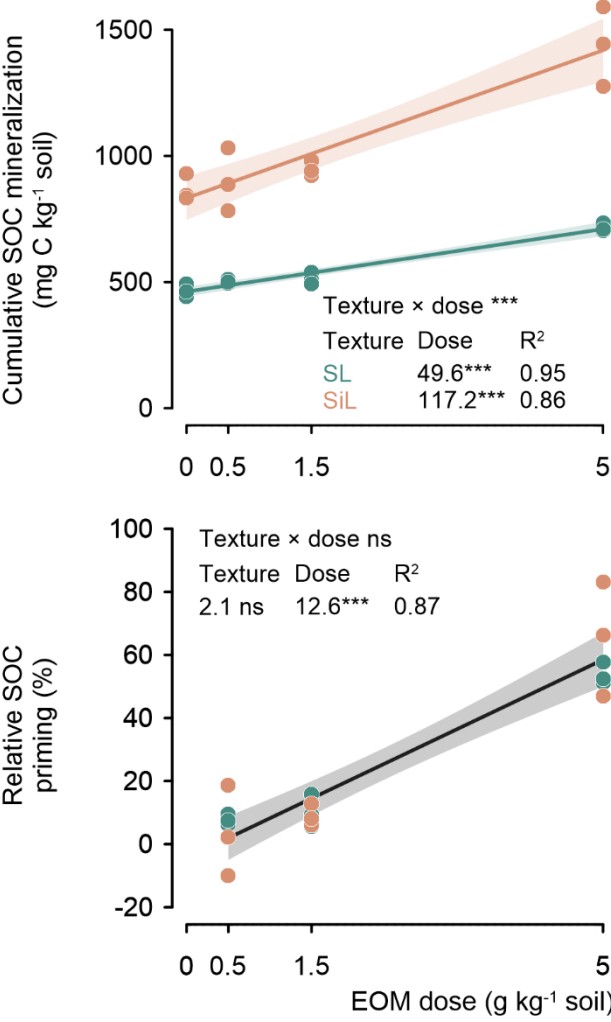

**Figure 3.** Cumulative mineralized native SOC (mg C kg⁻¹) after 90 days for a sandy loam (SL) and a silt loam (SiL) soil as a function of EOM (ryegrass) application dose (upper figure). The lower graph compares the extra amount of SOC mineralized vs. the unamended controls and relative to the unamended controls, i.e. EOM addition induced priming of native SOC in both soil textures. * denote

significance levels of the linear model interaction effect between soil texture and EOM dose, and for the linear response of cumulative and primed SOC mineralization to EOM dose, with *** <0.001, ** <0.01, * <0.05, and n.s. = not significant. Polygons around regression lines represent 95% confidence intervals.

### 3.3 Soil microbial biomass

MBC was linearly related to the EOM dose after 45 days of incubation ($P$ <0.001; Fig. 4 top). The increase in MBC in

response to increasing EOM dose was double in the silt loam compared with the sandy loam soil, i.e. +25.3 and +12.3 mg MBC per gram EOM added, respectively. The resulting overall increase in MBC was not proportional to the EOM dose; therefore, EOM-mediated MBC increase (i.e. the extra MBC in the EOM-amended vs. control soil) to the EOM dose tended to decrease with EOM dose, although not significantly ($P$ = 0.428; $R^2$ = 0.15; Fig. 4 bottom).

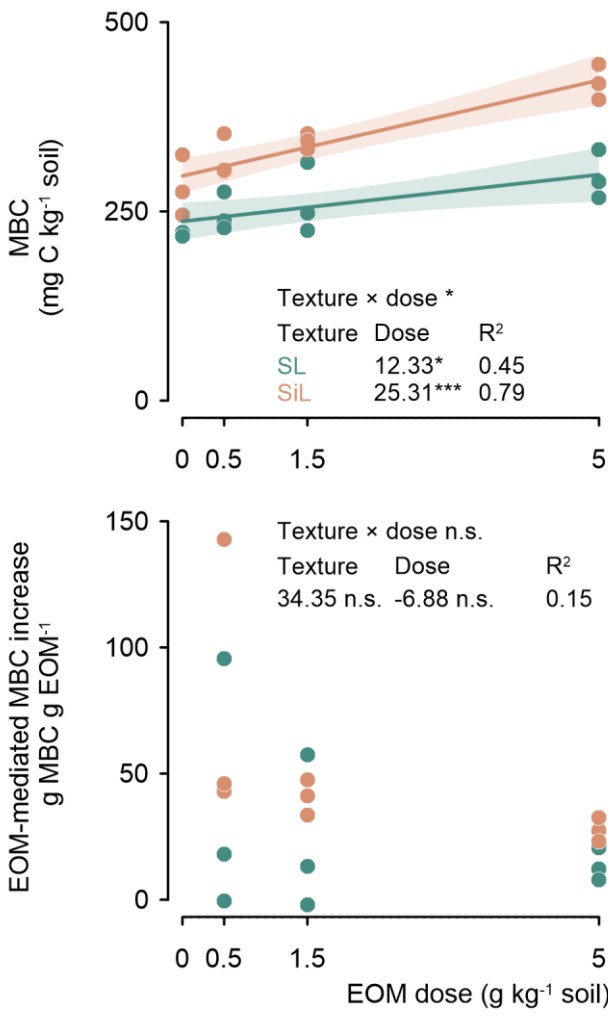

**Figure 4.** Microbial biomass carbon (MBC) after 45 days of incubation of the sandy loam (SL) and silt loam (SiL) soils amended with EOM (ryegrass) at various application doses (upper figure). The corresponding EOM-mediated MBC increase is presented in the lower figure. * denote significance levels effects of the linear model interaction between soil texture and EOM dose, and the effect of EOM dose on MBC and EOM-mediated MBC increase, with *** <0.001, ** <0.01, * <0.05, and n.s. = not significant. Polygons around regression lines represent 95% confidence intervals.

## 3.4 Soil redox potential

Soil Eh varied from approximately +550 to +850 mV in the sandy loam soil cores (Fig. 5**Figure**). It tended to decrease with increasing EOM dose until five days from the start of the incubation, after which it became indifferent with EOM addition; although in the control treatment, it mostly remained above the Eh in the EOM-amended soils. Nevertheless, no significant differences were observed in Eh between the EOM treatments in the sandy loam soil. In the silt loam soil, overall Eh was lower than that in the sandy loam soil and varied from approximately +470 to +700 mV. Although Eh in the silt loam soil was consistently lower for the 5 g kg$^{-1}$ soil EOM treatment during the first 41 days, the dose effect was not statistically significant, despite six replicate measurements per treatment. Overall, for both soils, the observed Eh ranges indicated prevailing oxic conditions (>300 mV), with $O_2$ being the main electron acceptor used in microbial respiration (Reddy and DeLaune, 2008). The non-significant interaction between dose × depth (5 and 15 cm below surface) for both soil textures demonstrated that the dosage effect on soil Eh did not differ among soil depths.

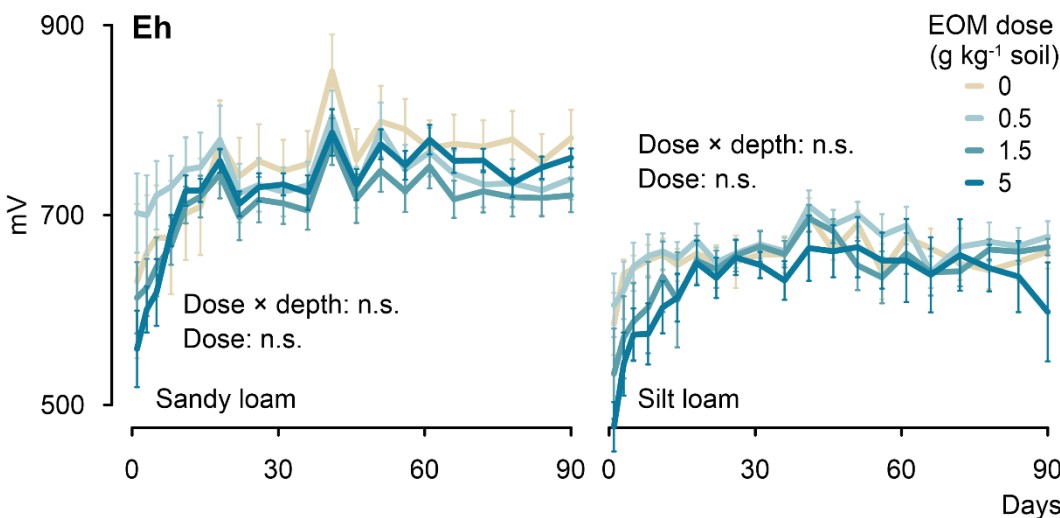

**Figure 5.** Evolution of the soil redox potential measured at 5 and 15 cm depth across the 90-day incubation experiment for the sandy loam and silt loam soils amended with different doses of EOM (ryegrass). Error bars show standard errors of means (n = 6).

### 3.5  Soil pore structure

In sandy loam soil, no effect of EOM addition was observed on the pore neck size distribution (Fig. 6). However, in case of the 5 g kg$^{-1}$ soil EOM dose, a reduction of the 3–9 μm class vol% by 20% compared with the unamended control ($P$ <0.05) was observed. In silt loam soil, EOM application more strongly affected the pore neck size distribution, especially at higher EOM doses. Adding EOM at 1.5 and 5 g kg$^{-1}$ soil increased the vol% of the >300 μm pore class by 48% and 49%, respectively, compared with the control (although only at $P$ <0.1). Addition of 5 g EOM kg$^{-1}$ also increased the volume fraction of the 60–100 μm class by 21% compared with the control ($P$ <0.05). The 43–60 μm class volume fraction was also larger in the 5 g kg$^{-1}$ soil EOM treatment than in the 0.5 g kg$^{-1}$ soil treatment ($P$ <0.05). EOM doses of 0.5, 1.5, and 5 g kg$^{-1}$ soil increased the volume fraction of the 3–9 μm class by 50%, 62%, and 32%, respectively ($P$ <0.05). In contrast, there was a decrease in the volume fraction of the 0.2–3 μm pore size class by 12%, 20%, and 17% with EOM doses of 0.5, 1.5, and 5 g kg$^{-1}$, respectively ($P$ <0.01).

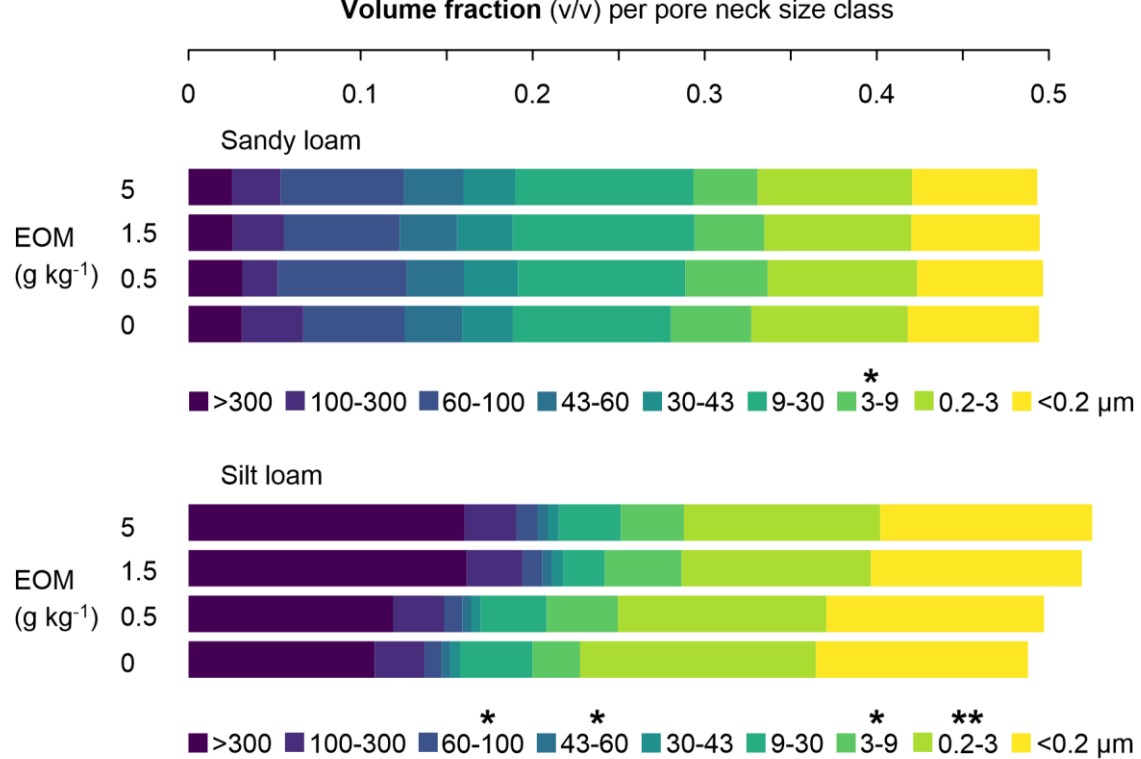

**Figure 6.** Volume fraction (unitless) of nine pore neck size classes (i.e. pores with given intervals of the pore neck diameter) in the sandy loam and silt loam soils after the application of EOM (ryegrass) doses and an unamended control (n = 4). * represent significant differences (one-way ANOVA) of the volume fraction of the pore neck class between EOM doses within each soil texture, with *** < 0.001, ** <0.01, * <0.05, and • <0.1.

### 3.6 Soil mineral nitrogen

Soil mineral N measured after 90 days of incubation increased linearly with EOM dose in both sandy loam and silt loam soils ($P$ <0.001; Fig. 6). It was always higher in the EOM-amended treatments than in the unamended controls. In the sandy loam soil, soil mineral N increased more strongly with EOM dose than in the silt loam soil, with values of +13.1 vs. +6.9 mg N per g EOM added, respectively.

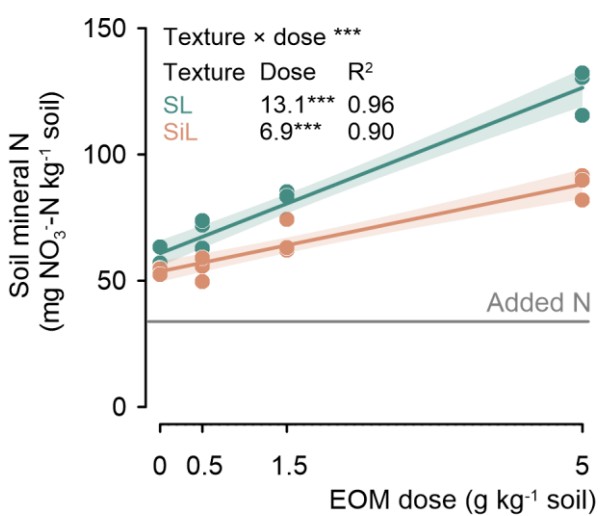

**Figure 7.** Soil mineral nitrogen ($NO_3^-$-N) content at the end of the 90-day incubation as a function of EOM dose for a sandy loam (SL) and a silt loam (SiL) soil. * denote significance levels of the linear model variables soil texture and EOM dose and their interaction on soil mineral N, with *** <0.001. Polygons around regression lines represent 95% confidence intervals.

## 4 Discussion

### 4.1 Mineralization of EOM as a function of its application dose

In the sandy loam soil, EOM-derived C mineralization was independent of its application dose, whereas it decreased exponentially with increasing EOM dose in the silt loam soil (Fig. 2), contradicting our hypothesis that the relative EOM mineralization would increase with increasing EOM dose. Overall, around 50 and 75% EOM-C was mineralized in the sandy loam and silt loam soils after the 90-day incubation experiment (Fig. 2). The ordination of relative EOM mineralization patterns remained consistent among the established dose treatments over time and are projected to remain likewise for at least some time (Fig. A1). For instance, mineralization over 137 days at the established 20°C in the lab experiment equates to about one year in the field in Belgium (9.7°C on average) (De Neve et al., 1996). Thus, the present analysis suggests that within the time scale of about one year we might expect no dosage response of EOM mineralization in the sandy loam soil and a negative response with dose for the silt loam soil. The unresponsiveness of relative EOM mineralization to EOM application dose in the sandy loam soil was consistent with the results of Shahbaz (2017a), who observed equal relative degradation of wheat crop residue with EOM dose of 1.40 and 5.04 g kg$^{-1}$. We previously also found the response of maize straw C mineralization to be unresponsive to its application dose in soils with contrasting native soil organic matter (SOM) levels (Mendoza et al., 2022a), and likewise for ryegrass (Mendoza et al., 2022b). These results are, however, in contrast to a study by Don et al. (2013), who reported that the mineralization of total soil C (SOC and EOM) was much slower when compost was diluted in a soil column than when it was concentrated. They postulated that closer proximity between the substrate and decomposers would allow more efficient decomposition. In our experiment, the EOM-mediated MBC increase was proportional to EOM dose in both soil textures (Fig. 4; bottom), suggesting that further growth of MBC was energetically equal for the different included substrate concentrations. This thus not support a view by Don et al. (2013), nor that by Ekschmitt et al. (2005), who postulated, based on the modelling output of an inverse Michaelis-Menten equation, that as the enzyme pool increases, the activity per unit enzyme decreases, with a lower relative C decomposition at higher C concentrations. They concluded that energy costs are not returned by decomposition products over a certain enzyme production rate, creating a negative feedback loop for microbial activity.

In the silt loam EOM mineralization was restricted with increasing EOM dose. First, with an abundance of labile C added, as was the case here, a shortage of mineral N could have limited microbial growth. However, such an effect could be excluded because high soil N concentrations (>50 mg $NO_3^-$-N kg$^{-1}$) were observed at the end of the experiment in all treatments, likely as a result of the initial large added dose of N (35.7 mg $NO_3^-$-N kg$^{-1}$) and net N mineralization from EOM, indicated by the proportional increase of final mineral N content to EOM dose (Fig. 7). Second, the increase in MBC and microbial respiration caused by high EOM doses could result in excessive $O_2$ demand that, when not met by $O_2$ diffusion could limit EOM degradation. Such $O_2$ limitations due to higher EOM doses may be expected to be more severe in finer-textured soil due to lower air permeability. Soil Eh was indeed generally lower in the silt loam soil than in the sandy loam soil, and increasing EOM dose more strongly decreased the soil Eh in silt loam soil, but still Eh remained at levels indicative of aerobic conditions (Fiedler et al., 2007; Husson, 2013). Furthermore, as $NO_3^-$ concentrations increased with increasing EOM dose, there was no indication of more denitrification with increasing EOM dose. The overall oxidised soil state, even in the finer-textured soil with a high dose of EOM, might be linked to the development of the soil pore network, namely, high EOM doses significantly increased the vol% of larger pore neck classes (i.e. 43–60, 60–100, and >300 μm) in the silt loam soil. Hence, from the Eh readings, we observed no indication that $O_2$ limitations

would have caused the lower relative EOM mineralization in the silt loam soil at higher EOM doses. In conclusion, we could not identify the cause of these phenomenon, and further research is required to explore the potential mechanisms leading to a relative temporal stabilization of EOM when added at larger doses. For example, occlusion of EOM in soil aggregates is known to provide some degree of physical protection against decomposition, but the effect of EOM dose thereupon was not assessed in this study. Soil aggregate formation is stimulated by OM addition, and De Gryze et al. (2005) reported greater development of macroaggregates at higher EOM doses. Therefore, it would be reasonable to postulate and further investigate that physical protection of EOM by aggregate occlusion is favoured at higher application doses, causing negative feedback on the relative EOM mineralization.

Drawing conclusions for EOM management in the field based on this 90-day lab incubation experiment at 20°C is to be made with care. Nevertheless, well aligned temporal courses of the cumulative EOM mineralization (Fig. A1) suggest at least on the short term an unchanged ordination of further relative EOM mineralization between the dose treatments. When extrapolated to 137 days (not shown), more or less equivalent to C mineralization occurring in one year in the field at 9.7°C in Flanders (De Neve et al., 1996), our results suggest no or but a limited negative effect of adding EOM at increasing doses on its annual mineralization, a traditionally used metric in C-balance calculations (the so-termed humification coefficient). However, empirical evidence from field experiments is now needed to confirm these findings.

### 4.2 Effect of EOM application dose on native SOC mineralization

We hypothesized that SOC mineralization would be stimulated by increasing the EOM dose with stronger effects in sandy loam soil, as SOC may be less stabilized than in silt loam soil. Priming of SOC mineralization was linearly related to the ryegrass application dose ($P$ <0.001), with native SOC mineralization stimulated by 50 and 125 mg C per gram of EOM added in the sandy loam and silt loam soils, respectively ($P$ <0.001). Shahbaz et al. (2017a) reported that addition of ground wheat residue of 1.40 and 5.04 g kg$^{-1}$ soil to a silt loam soil increased SOC mineralization by 50% and 90%, respectively. In our study, SOC mineralization increased by 8%, 10%, and 54% in the sandy loam soil, and by 4%, 9%, and 65% in the silt loam soil from the low to high EOM doses. In agreement with our study, a positive linear relationship between glucose dose (0.008 to 1.606 g C kg$^{-1}$ week$^{-1}$) and SOC priming was reported by Liu et al. (2017). In contrast, Xiao et al. (2015) reported a decrease in the priming of SOC per unit of litter (mix of <2 mm aboveground plants of a steppe vegetation) added (0, 60, 120, 240, and 480 g C m$^{-2}$). Moreover, Guenet et al. (2010) reported that addition of wheat straw (3.5, 5.2 and 7.5 g kg$^{-1}$ soil) did not proportionally stimulate SOC mineralization. These outcomes are in contrast with the observed proportional priming of SOC with increasing EOM in this study. Nitrogen availability did not likely limit heterotrophic activity in our study, but it possibly did restrict C mineralization at increasing EOM doses in the experiments of Guenet et al. (2010) and Xiao et al. (2015). A commonly proposed mechanism to explain SOC priming is the 'N-mining theory' (Craine et al., 2007), which assumes that microbes decompose native SOC in search of mineral N to meet their metabolic demands. As explained above, with extra mineral N added initially to our soils and a clear further net soil N mineralization, this mechanism can largely be ruled out. Alternatively, SOC priming is often explained by the 'co-metabolism hypothesis (Bingeman et al., 1953; Kuzyakov and Domanski, 2000), i.e. application of a labile substrate, such as the ryegrass used here, stimulates microbial biomass growth and enzyme production, which in addition to decomposing EOM, also triggers native SOC mineralization. MBC was linearly related to the EOM dose in both soils (Fig. 4) and positively correlated with the rate of relative SOC priming in both sandy loam and silt loam soils (r = 0.63 and 0.72, respectively; $P$ <0.05). While these trends support the idea that increased microbial biomass and activity with EOM addition primed SOC mineralization, further proof is required to identify co-metabolism as the principal mode of SOC priming. Most studies (e.g. Xiao et al. (2015)) have shown that priming of SOM mineralization relates linearly to MBC but indeed likewise could not unequivocally pinpoint the mechanisms involved. An overall stronger priming of

SOC per g soil in the silt loam soil, regardless of EOM dose, likely results from an inherent better SOC degradability compared to the sandy loam soil, as evidenced by the much higher MBC and SOC mineralization in the unamended control soils (Fig. 3 and Fig. 4). This contrast in MBC and heterotrophic activity is unlikely to have been the result of differences in SOC content, content of pedogenic oxides or pH, as these properties were very similar between both textured soils. Instead, this more likely results from a combination of differences in SOC quality and soil physical structure between both textures, but such effects are difficult to discriminate from one another.

The duration of the present 90-day incubation experiment was sufficiently long to capture trends in SOC priming. The temporal course of the SOC priming response to EOM dose was also similar in both textured soils. In sandy loam soil, increasing EOM dose induced higher positive SOC priming between days 2 and 12 of the incubation (1.5 g kg$^{-1}$ soil vs. control, $P = 0.03$; 5 g kg$^{-1}$ soil vs. control, $P < 0.001$), whereas priming was not significant from day 12 on. In silt loam soil, SOC priming occurred between days 2 and 17 of the incubation (5 g kg$^{-1}$ soil vs. control, $P = 0.001$) but no longer thereafter (Fig. B1). As SOC priming was thus rather short-lived, this allows careful projections for the relevance of EOM dose responses on priming in the overall SOC balance, which considers EOM and SOC mineralized as compared to added and initially present organic C, respectively. This balance proved negative or close to zero for the silt loam soil and the low EOM dose in the sandy loam soil (Table C1). In other words, C mineralization exceeded the added C dose with eventual net loss of C relative to initially present as SOC and added via EOM. This result is remarkable at first sight but is explained by the very high observed SOC mineralization of 3-9% of SOC, a situation typical for lab incubation studies that is not met in the field, which in the area where soil was collected would amount to 1-3% of SOC being mineralized across the course of an entire year. Hence, we may only relatively compare the net C balance of the used EOM dose scenarios in this study. The net C balance was also suggested to differ with EOM dose in both soils, it was in particular lower when EOM was added at low dose than when no EOM was added at all. The slowed relative EOM mineralization in the silt loam soil that was not seen in the sandy loam soil did not result in an increased net C balance as its impact was counteracted by a relatively stronger SOC priming response to EOM dose in the silt loam than in the sandy loam soil. Priming contributed from around 4% up to 40% of to this overall large SOC mineralization going from low to high application doses in both soils (Table C1) and was thus certainly a non-negligible term in the net C balance. The contrast in C balance per unit of EOM dose followed the order: intermediate > high > low doses, we thus might tentatively conclude that adding EOM at low dose is unfavorable from a C balance perspective. Finally, it is important to note that formation of new SOC from decomposing EOM was not considered here, although this would yield a more accurate prediction of the net effect of EOM doses on the SOC balance than when using C mineralization data. Our experimental set-up did not allow the detection of remnant EOM and newly formed SOC against the native SOC background. Although not confirmed statistically, it is noteworthy that, especially for silt loam soil, less MBC was produced per unit of EOM added. Microbial biomass-derived necromass contributes to SOC formation (Kästner et al., 2021) and according to Liang et al. (2019) can make up to 56% of the total SOC in temperate agricultural soils. A relative decrease in the formed MBC per unit of EOM added might thus adversely impact SOC preservation when EOM is added at fewer but larger doses. Ultimately, long-term field observations are again required to confirm the impact of EOM dosage on SOC storage.

### 4.3 Effect of soil pore structure on priming of SOC mineralization

We also investigated whether EOM application dose affected soil structure and if this in turn explained the priming of SOC mineralization. We did not observe major changes in the soil structure with respect to different EOM doses in the sandy loam soil. However, EOM stimulated the volume percentage of several larger pore classes (43–60, 60–100 and >300 μm; $P < 0.05$, 0.05, and 0.1, respectively) in the silt loam soil. Even though the relative SOC priming did not differ among soil textures, the slope of the relative SOC priming increased by 16% in silt loam soil compared to 11% per g

EOM added in sandy loam soil. We therefore hypothesize that the development of macroporosity might have contributed to the promotion of relative SOC priming in the silt loam soil. In fact, there were positive linear relationships (via linear regressions) between the silt loam soil volume fraction of pore neck size classes 60–100 and >300 μm and relative SOC priming ($R^2 = 0.34$ and 0.36; and, $P = 0.09$ and 0.08, respectively), and a negative relation with the 3-9 μm class that also depended on EOM dose, whereas no such relationships existed for the sandy loam soil. The contrasting unresponsiveness of EOM mineralization to EOM dose as compared to SOC priming could be explained as follows. In particular, the majority of SOC is usually mineral-associated in agricultural soils (Kögel-Knabner et al., 2008), and is therefore situated within small pores where oxygen provision can be readily constrained (Kuka et al., 2007), while the added discrete substrate particles necessarily reside in large macropores, due to which SOC and not EOM mineralization may sooner become $O_2$ limited. Mineralization of SOC would therefore be logically more dependent on moderation of soil structure towards more macropores caused by EOM amendment. Experimental verification of this hypothesis will be challenging as spatial mapping of $O_2$ availability in soil pore space is practically difficult. As a first step, modelling of soil $O_2$ transport in 3D soil models based on soil pore models derived from CT volumes could provide further insights in the link between SOC mineralization and pore structure (Schlüter et al., 2022), but such an approach then needs to be validated, e.g. through integration of $O_2$ microsensors with pore metrics derived from X-Ray μCT, as recently demonstrated by Rohe et al. (2021). We further compared the results with our previous study using the same soils (although sampled at different times in the field) and exactly the same source of EOM (although more refined) as used here. In the previous study, ryegrass doses of 0.5 and 5 g kg$^{-1}$ soil stimulated macroporosity formation and SOC mineralization by 30% and 71% in sandy loam soil, but only by 28% in silt loam soil at high dose when the soil structure was apparently unaffected (Mendoza et al., 2022b). Due to differences in initial soil disturbance, in the current experiment, the silt loam soil had a much larger fraction of very small pores (>10% of the soil volume consisted of <0.2 μm pores) when compared to that in (Mendoza et al., 2022b). Improved soil aeration, which likely resulted from the increase in the vol% of >300, 60–100, and 43–60 μm pore classes with EOM added, could thus again explain the stronger observed stimulation of SOC mineralization in this study, when compared to Mendoza et al. (2022b). Interestingly, the soil Eh data did not indicate improved aeration in the silt loam soil at higher EOM application doses, but this was not necessarily expected because the two phenomena that co-determine Eh probably could have counterbalanced each other as follows: improved aeration and a corresponding increase in soil Eh with more EOM added vs. at the same time also more heterotrophic activity (and corresponding electron donation). A more systematic approach combining microscale soil Eh measurements, soil pore network structure and soil respiration, would enable confirmation of this potentially interesting link between Eh and porosity, and their resulting effect on SOC dynamics . An overview of the main mechanisms by which EOM application dose affects the relative EOM and native SOC mineralization is presented in Fig. 8.

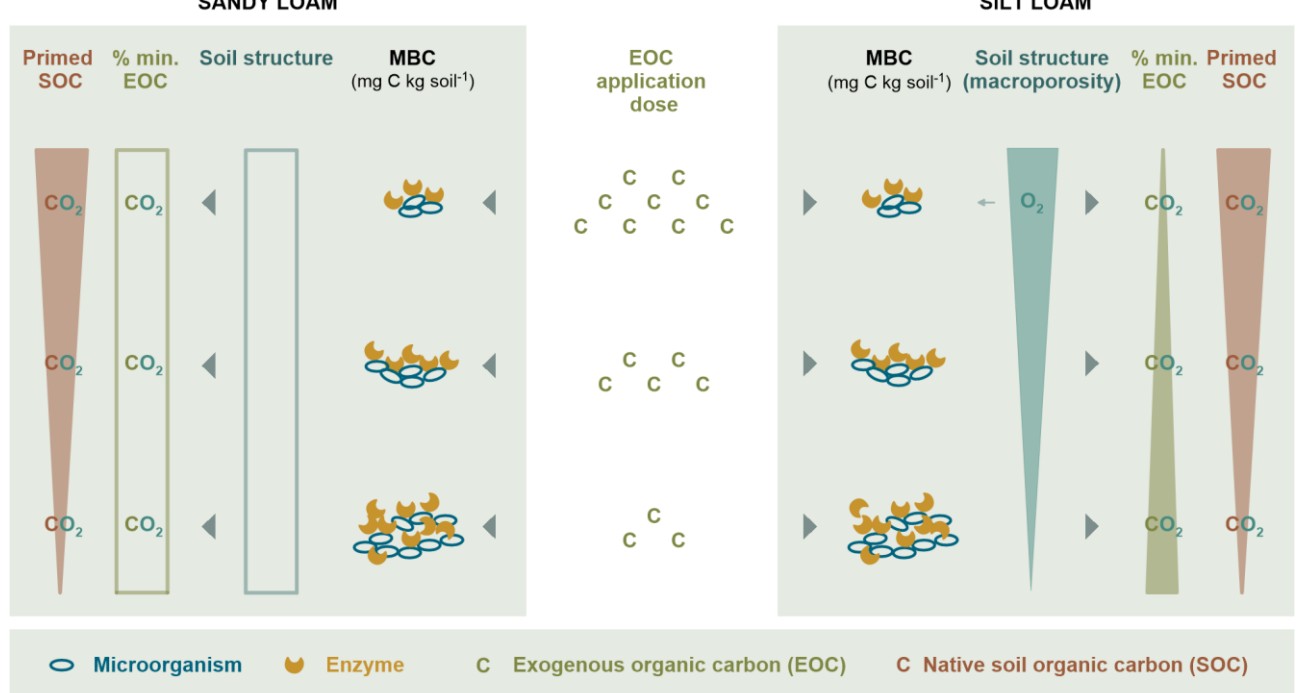

**Figure 8.** Overview of the mechanisms shaping the relative EOC mineralization (% min. EOC) and SOC priming effects in response to EOM application dose. In the sandy loam soil (left-hand side), increasing EOM dose supplied energy for microbial growth and extracellular enzyme production, which possibly degraded native SOC (i.e. co-metabolism), but did not affect the mineralized EOM fraction. Soil Eh decreased slightly with increasing EOM dose, but C mineralization remained aerobic as expected in the sandy loam soil. Here, soil structure was not much affected by EOM dose. In the silt loam soil (right-hand side), where $O_2$ diffusion is expected to be inherently constrained, increasing EOM dose induced macroporosity which also compensated the large $O_2$ consumption due to larger microbial growth. Here, mainly co-metabolism could explain the positive priming effects with increasing EOM dose in the silt loam soil. The decreasing percentage of decomposed EOM with increasing EOM application dose observed in the silt loam soil might be related to larger EOM protection within aggregates, but further confirmation of this hypothesis is required.

## 5 Conclusion

Limited research exists on the effect of EOM dose on C mineralization in soil, and consequently, this effect has also been largely overlooked by SOC simulation models. Our results showed a dose response of relative EOM mineralization in heavy- but not light-textured soil. The formation of MBC was independent of EOM dose; thus, we found no evidence suggesting a more economical growth of heterotrophs at higher substrate doses. We expect that with the generally observed lower bulk soil Eh in the silt loam soil, the slowed relative mineralization of EOM at increasing dose could be related to enhanced occurrence of local $O_2$ limitation surrounding EOM litter, even though its addition in fact also stimulated macroporosity. Revealing causality and identifying the situations where increased $O_2$ demand due to enhanced microbial activity at higher EOM dose outweighs the potentially improved gaseous transport from increased macroporosity will require experiments targeting changes in soil structure and include local Eh measurements at the microscale. Tentative C balance calculations finally pointed out that when EOM is added at a low dose, i.e. around 0.5 g kg$^{-1}$, it has a negative impact on SOC. When added at 1.5 or 5 g kg$^{-1}$ a positive effect is expected on the C balance. However, these tentative C balances of this upscale pot experiment should now be confirmed in the field where environmental conditions vary.

**Appendix A**

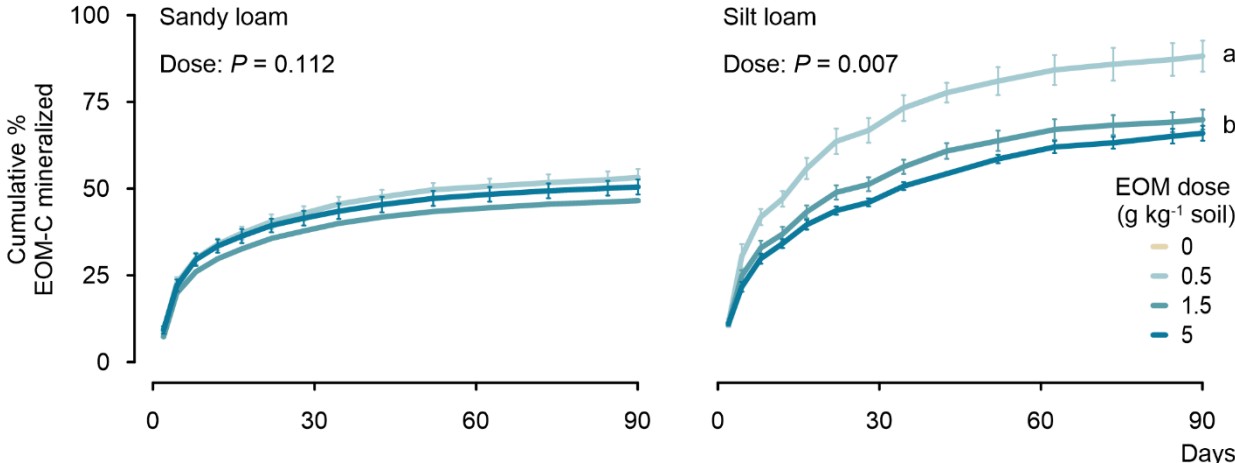

**Fig. A1.** Cumulative % EOM-C mineralized over the 90-day incubation experiment for a sandy loam and silt loam soils as a function of application dose. *P* values indicate the effect of application dose on the percentage of EOM mineralized at the end of the experiment. Error bars show standard errors of means (n = 3).

**Appendix B**

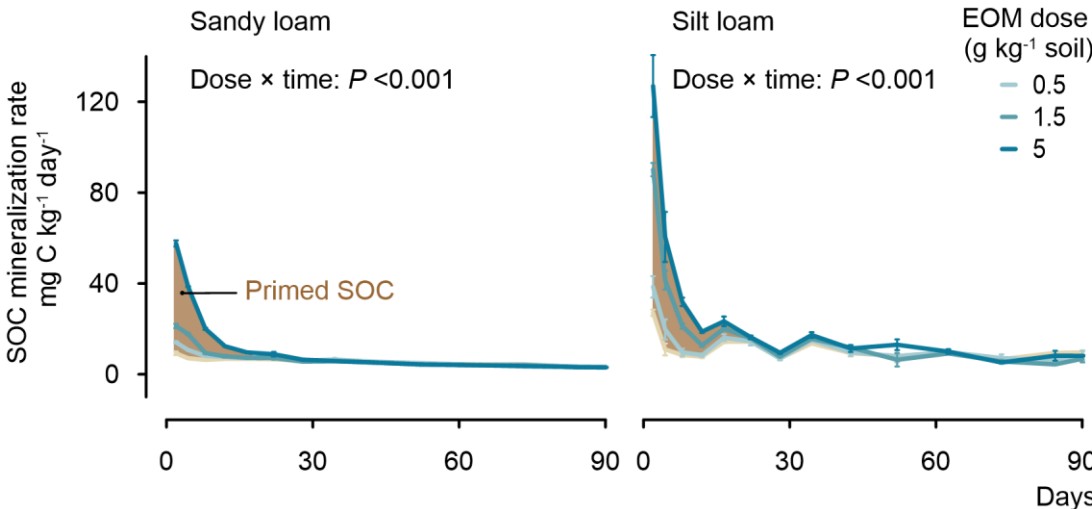

**Fig. B1.** The SOC mineralization rates over the 90-day incubation experiment for a sandy loam and a silt loam soil in response to EOM application dose. Shadow brownish areas depict priming of SOC mineralization caused by EOM application dose. In sandy loam soil, EOM dose induced SOC priming between days 2 and 12 of the incubation (*P* values for 1.5 g kg$^{-1}$ vs. control = 0.03; 5 g kg$^{-1}$ vs. control <0.001; 5 vs. 0.5 g kg$^{-1}$ <0.001; and, 5 vs 1.5 g kg$^{-1}$ <0.001), whereas priming remained not significant during day 12 to 90. In silt loam soil, EOM dose induced SOC priming between days 2 and 17 of the incubation (*P* values for 5 g kg$^{-1}$ vs. control = 0.001; 5 vs. 0.5 g kg$^{-1}$ = 0.002; and, 5 vs 1.5 g kg$^{-1}$ = 0.01), while priming was not significantly different between EOM doses from day 17 to 90. Error bars show standard errors of means (n = 3).

**Appendix C**

**Table C1.** Soil carbon balance between SOC present initially and EOM-C input and outputs (mineralization of EOM and SOC) in the two soils used in the incubation experiment. Averages of three replicates with standard errors are shown.

|  | **Sandy loam** | | | **Silt loam** | | |
|---|---|---|---|---|---|---|
| **EOM dose (g DM kg soil)** | 0.5 | 1.5 | 5 | 0.5 | 1.5 | 5 |
| **EOM-C (mg C kg soil)** | 202.2 | 606.6 | 2022 | 202.2 | 606.6 | 2022 |

| | | | | | | |
|---|---|---|---|---|---|---|
| Mineralized EOM-C (mg $CO_2$-C kg soil) | 117.0±5.3 | 306.8±2.3 | 1109.4±47.4 | 193.5±9.9 | 459.6±18.9 | 1445.8±47.1 |
| Initial SOC (mg C kg soil) | 14400 | 14400 | 14400 | 14900 | 14900 | 14900 |
| Mineralized SOC (mg $CO_2$-C kg soil) | 501.4±4.4 | 513.3±13.8 | 715.6±9.3 | 900.3±30.6 | 946.8±72.3 | 1436.6±17.0 |
| Priming contribution to mineralized SOC (%) | 7.2±0.9 | 9.3±2.7 | 35.0±1.3 | 3.5±8.0 | 8.3±1.8 | 39.5±6.3 |
| C balance end (mg C kg soil) | 13983.8±6.0 | 14186.5±13.1 | 14597.0±42.6 | 14008.5±82.0 | 14100.2±34.7 | 14039.6±113.6 |
| C balance end in controls (mg C kg soil) | 13934.6±14.5 | 13934.6±14.5 | 13934.6±14.5 | 14031.4±31.6 | 14031.4±31.6 | 14031.4±31.6 |
| C balance end vs. C balance end controls (mg C kg soil) | 49.1±6.0 | 251.9±13.1 | 662.4±42.6 | -22.9±82.0 | 68.8±34.7 | 8.2±113.6 |
| C Δ balance per unit EOM added (Δ mg C mg EOM-C$^{-1}$) | 0.24±0.03 | 0.42±0.02 | 0.33±0.02 | -0.11±0.41 | 0.11±0.06 | 0.00±0.06 |

**Data availability.** The data generated in this study are available from the corresponding author upon reasonable request.

**Author contributions.** OM, SDN and SS conceptualized the study and acquired funding. OM, HD, HL, and AF performed the experiment. OM wrote the original draft, and all the authors edited and reviewed the manuscript.

**Conflicts of interests.** At least one of the (co-)authors is a member of the editorial board of SOIL.

**Acknowledgments.** This work was supported by the Research Foundation Flanders (FWO: G066020N). We acknowledge the Secretaría Nacional de Educación Superior, Ciencia, Tecnología e Innovación (Senescyt-Ecuador) for funding Orly Mendoza. We also thank Mathieu Schatteman, Tina Coddens, Anne-Mie Terryn, Sophie Schepens, Maarten Volckaert, and Thu Tran for their assistance during chemical and physical analyses and for setting up the experiments.

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
