# Peer review of "Soil organic carbon mineralization is controlled by the application dose of exogenous organic matter"

_EGUsphere, 2024_

## Author Response (AR1)

Dear editors and referees,

We hereby present a point-by-point reply to the SOIL reviewers' and editor's comments of our submitted manuscript, EGUSPHERE-2024-107. All suggested changes mentioned in our previous response letter, uploaded in the online discussion on June 9, 2024, have been incorporated.

- All reviewer's comments have been addressed and are presented below in blue.
- Literal suggested excerpts added to the revised manuscript are in italics and quotes.
- Changes made in the document are reported by line number in the manuscript, with tracked changes still visible.
- All figure numbers have been updated, as one figure was included in the introduction as suggested by the reviewers. Other figures have also been updated following the reviewers' suggestions.

We thank the reviewers and topic editor for their constructive feedback which has significantly improved the analysis and quality of our manuscript. We hope that the changes made are satisfactory and meet the criteria for publication in SOIL.

Kind regards,

The authors

**Editor**

The request to highlight the novelty of our manuscript in relation to a manuscript published in Biology and Fertility of Soils has been addressed in the introduction (L.88-97). Based on reviewer #2's suggestions, we have included discussions on relative SOC priming and MBC, and have interpreted the results accordingly. All changes are explicitly detailed below. Further justification of the statistical analysis has been provided in the respective section (L.230), and minor changes have been implemented. All suggested changes are thoroughly addressed below.

**Reviewer #1:**

The authors present the results of a laboratory study, in which the mineralization patterns of plant litter is investigated in two soils of different texture. The aim of the study is to investigate how different doses of plant litter may influence the mineralization rate and priming of the initial SOC.

The study is nicely written, easy to read and follow. The experiment itself is interesting and fits into the scope of the journal. However, and I am really sorry to say this, I see several major critic points that make the manuscript unsuitable for publication in my opinion:

1. The authors investigate the mineralization of added OC and initial SOC and distinguish between them with a 13C label. This is valid to do so. However, I could not find any data on the native C concentration in the soil and the changes therein. It seems to me, that only respired C has been measured. By this, a calculation of the overall net C balance in the soils is impossible (and has not been done by the authors). This means that it is impossible to detect any C losses other than

as CO2 (e.g. by methane, leaching, etc.). The priming effect seems to have quite an influence, but the supplementary materials show, that it is quite limited in time. The authors should therefore provide a clear balance (table, etc.) showing how much overall C can be added to the SOC by the various C inputs and whether the overall balance would be positive or negative. It is really necessary to show, how much of the overall external carbon ended up in the soil and how much the priming effect influenced the overall carbon storage in the soil. The fate of added OC and whether it is causing priming or SOC formation is currently a hot topic at the moment, although there is a big debate about its impact on the net C balance. Therefore, without this data I cannot see how the presented study adds any novel aspect to the current scientific knowledge.

We understand the reviewer's comment in the sense that a balance between C inputs and outputs should be presented. We have now done this and added the inputs (C added with exogenous organic matter), initial native SOC contents, and mineralization of EOM-C and native SOC. We have also included the relative contribution of priming to overall native SOC mineralization and calculated the C balance. These results are presented in Table C1 in the supplementary materials (L.560) and a discussion has been modified/added as follows (L.438-453): *"As SOC priming was thus rather short-lived, this allows careful projections for the relevance of EOM dose responses on priming in the overall SOC balance, which considers EOM and SOC mineralized as compared to added and initially present, respectively. This balance proved negative or close to zero for the silt loam soil and the low EOM dose in the sandy loam soil. In other words, C mineralization exceeded the added C dose with eventual net loss of C vs. initially present as SOC and added via EOM. This result is remarkable at first sight but is explained by the very high observed SOC mineralization of 3-9% of SOC, a situation typical for lab incubation studies that is not met in the field, which in the area where soil was collected would amount to 1-3% of SOC being mineralized across the course of an entire year. Hence, we may only relatively compare the net C balance of the here used EOM dose scenarios. The net C balance was also suggested to differ with EOM dose in both soils, it was in particular lower when EOM was added at low dose than when no EOM was added at all. The slowed relative EOM mineralization in the silt loam soil that was not seen in the sandy loam soil did not result in an improved net C balance as its impact was counteracted by a relatively stronger SOC priming response to EOM dose in the silt loam than in the sandy loam soil. Priming contributed from around 4% up to 40% of to this overall large SOC mineralization going from low to high application doses in both soils (Table C1) and was thus certainly a non-negligible term in the net C balance. The contrast in C balance per unit of EOM dose followed the order: intermediate > high > low doses, we thus might tentatively conclude that adding EOM at low dose is unfavorable from a C balance perspective."*

**Table C1.** Soil carbon balance between SOC present initially and EOM-C input and outputs (mineralization of EOM and SOC) in the two soils used in the incubation experiment. Averages of three replicates with standard errors are shown.

| | Sandy loam | | | Silt loam | | |
|---|---|---|---|---|---|---|
| EOM dose (g DM kg soil) | 0.5 | 1.5 | 5 | 0.5 | 1.5 | 5 |
| EOM-C (mg C kg soil) | 202.2 | 606.6 | 2022 | 202.2 | 606.6 | 2022 |
| Mineralized EOM-C (mg $CO_2$-C kg soil) | 117.0±5.3 | 306.8±2.3 | 1109.4±47.4 | 193.5±9.9 | 459.6±18.9 | 1445.8±47.1 |
| Initial SOC (mg C kg soil) | 14400 | 14400 | 14400 | 14900 | 14900 | 14900 |
| Mineralized SOC (mg $CO_2$-C kg soil) | 501.4±4.4 | 513.3±13.8 | 715.6±9.3 | 900.3±30.6 | 946.8±72.3 | 1436.6±17.0 |
| Priming contribution to mineralized SOC (%) | 7.2±0.9 | 9.3±2.7 | 35.0±1.3 | 3.5±8.0 | 8.3±1.8 | 39.5±6.3 |
| C balance end (mg C kg soil) | 13983.8±6.0 | 14186.5±13.1 | 14597.0±42.6 | 14008.5±82.0 | 14100.2±34.7 | 14039.6±113.6 |
| C balance end in controls (mg C kg soil) | 13934.6±14.5 | 13934.6±14.5 | 13934.6±14.5 | 14031.4±31.6 | 14031.4±31.6 | 14031.4±31.6 |
| C balance end vs. C balance end controls (mg C kg soil) | 49.1±6.0 | 251.9±13.1 | 662.4±42.6 | -22.9±82.0 | 68.8±34.7 | 8.2±113.6 |
| C Δ balance per unit EOM added (Δ mg C mg EOM-C$^{-1}$) | 0.24±0.03 | 0.42±0.02 | 0.33±0.02 | -0.11±0.41 | 0.11±0.06 | 0.00±0.06 |

We did not measure SOC contents at the end of the incubation period, because it is simply impossible to do this with sufficient accuracy to detect (small) changes in SOC in short term experiments. We estimated the net C balance at the end of the experiment using the C inputs and outputs. Leaching losses of OC in this experiment can be excluded given that our setup was sealed at the bottom. C losses by methanogenesis can be excluded given the relatively low water-filled pore space i.e. 55%, and the resulting aerobic conditions (redox potential >300 mV, measured throughout the incubation period in both soils). Supporting this, we even discarded denitrification as a main process not only because of the oxic redox potentials observed in our experiments but also because of the high nitrate concentrations with high application doses still present at the end of the incubation period (L.340 of the previous manuscript or L.382 in the present manuscript).

2. There is a quite similar study published by the same authors in the journal Biol. Fertil. Soils (2022). According to my understanding, this ealier study uses the same plant species and the same soils. The only difference I could find is the author's explanation that the previous study was "sampled at different times in the field?" (L.426) and there were two levels of disturbance. It seems to me that this previous study covers the same aspects than the presented study, but is much more comprehensive and better explained. I am sorry to say, but I cannot really see the novel aspect of the presented manuscript. It seems to be rather a subchapter or preliminary study to the already published study.

We respectfully disagree with the comment concerning the novelty of the study in relation to a previous study published in Biology and Fertility of Soils. Our previously published study focused on soil structure formation as created by soil sieving and application dose. The scale of that experiment was small. The present manuscript wanted to explicitly test how non-refined plant residues and much larger soil masses (we used 6.5 to 7 kg dry soil representing 23 times larger soil mass as compared to our previous study) influence C mineralization. Indeed, the contact between plant residues and soil was hypothesized to have stronger effects on the formation of C-hotspots caused by the residue chunks (instead of more homogeneous distribution of finely chopped residues as in the previous published work) causing potentially a local limitation of $O_2$ which could affect EOM and SOC mineralization. Following the results of our previous work, in the present manuscript, we thus attempted to create conditions that resemble the situation in the field, in order to gain a more realistic understanding of the application dose effect on the mineralization of exogenous organic matter and native SOC. We also refer to the appreciation by the second reviewer of the novelty and potential impact of our study. However, we do recognize that the novelty could have been better introduced, and we have now further explicitly clarified this novelty in the Introduction section (L.88-97) as follows: *"To account for these shortcomings and to obtain more realistic understanding of the application dose effect on EOM and SOC mineralization as it might occur in the field, we aimed to study decomposition of large pieces of $^{13}C$-labelled ryegrass residue in large relatively less disturbed soil cores. Consequently, as compared to Mendoza et al. (2022b) and Shahbaz (2017a), the soil masses used in a newly designed soil incubation experiment were about 23 times and 70 times larger, the soils were only coarsely sieved (<10 mm), the added crop residues were not chopped. Concentration of added ryegrass dose to non-refined ryegrass pieces is expected to yield a stronger local impact on soil structure and the concentration of C mineralization into such relative hotspots might more readily result in local depletion of $O_2$ with potential negative impacts on EOM and SOC mineralization as compared to previous work on this topic."*.

3. There is no information about the used soils and vegetation. The soil type, current vegetation or crops, and soil management, farming history (or at least the present use) should be provided. (The authors want to publish in a journal called "SOIL".) Without this information, the discussion section cannot be evaluated, especially not the part about initial SOC quality and soil structure. It is not enough to give only soil texture, pH and some total elemental concentration. In addition, there is no information on the C input by the ryegrass. Is this input realistic for those soils under the respective management? Has other literature suggested that the concentration will induce priming in similar soils? Also, how big where the ryegrass pieces? Could they be incorporated in aggregates?

We agree that the additional information required by the reviewer is important for the overall evaluation of the results. The soil type, current and past crops have now been added to table 1 of the manuscript (L.127).

Table 1. Physicochemical characteristics, and current and past crops of the two selected topsoils used in the soil incubation experiment. Averages of three replicates with standard errors are shown for pH, and measurements of two replicates are shown for Alox, Feox, P, Ca, K, and Mg.

| Soil characteristic | Sandy loam | Silt loam |
| --- | --- | --- |
| Sand (%) | 60.2 | 13.3 |
| Silt (%) | 26.9 | 67.0 |
| Clay (%) | 12.9 | 19.7 |
| SOC (g kg$^{-1}$) | 14.4±0.4 | 14.9±0.3 |
| $\delta^{13}$C (‰) | −27.72 | −25.11 |
| pH$_{(H2O)}$[a] | 6.4±0.0 | 7.4±0.0 |
| Al$_{ox}$ (mg kg$^{-1}$)[b] | 1199.4 – 1172.7 | 679.7 – 693.7 |
| Fe$_{ox}$ (mg kg$^{-1}$)[b] | 3221.0 – 3183.0 | 4280.5 – 4302.0 |
| P (mg kg$^{-1}$)[c] | 85.7 – 87.8 | 325.6 – 329.0 |
| Ca (mg kg$^{-1}$)[c] | 954.5 – 953.8 | 2429.2 – 2488.0 |
| K (mg kg$^{-1}$)[c] | 148.0 – 147.5 | 320.5 – 327.2 |
| Mg (mg kg$^{-1}$)[c] | 125.2 – 125.0 | 142.4 – 146.3 |
| Soil type | Cambisol | Cambisol |
| Current crop | French marigolds (*Tagetes patula* L.) | Maize (*Zea mays* L.) |
| Cropping history | 2012-2019: ryegrass | 2012-2019: potato, wheat, barley, maize, sugar beets, potato, wheat |

[a]pH$_{(H2O)}$ was obtained by inserting a glass pH electrode in 1:6.25 soil-H$_2$O mixture. [b]NH$_4$-oxalate extraction and detection using inductively coupled plasma optical emission spectroscopy (ICP-OES). [c]NH$_4$–acetate-EDTA extraction and ICP-OES analysis.

SOC quality might have some effect on structure formation, however, digging a bit deeper into the effect of SOC quality on structure formation would require a dedicated experimental setup with more than just two soils and this also moves us a bit too far from our hypotheses. In the submitted manuscript, we already mentioned that differences in soil priming across both soils "more likely results from a combination of differences in SOC quality and soil physical structure between both textures, but such effects are difficult to discriminate from one another." (L.431-432). In lines 101-102 of the previous version of the manuscript (L.122-124 in the current manuscript), we already included the C content of the ryegrass: "After 8 weeks, ryegrass was cut with a sickle and had a resulting $\delta^{13}$C of +51±5.4 ‰ (n = 6), C content of 439.3±5.4 g kg$^{-1}$, and C:N of 10±0.5". Also, in lines 114-116 (L.136-138 in the current manuscript) we previously already included the following sentence "These applications were used as representative for rather low, intermediate, and high EOM doses commonly applied in agricultural practices under field conditions, and they correspond to 1.75, 5.25, and 17.5 Mg ha$^{-1}$ assuming a depth of 25 cm and a bulk density of 1.4 g cm$^{-3}$." Concerning the reviewer's question to report on similar "dose studies" and priming of native SOC

mineralization; we did already discuss such studies (e.g. Shahbaz et al. 2017a, Liu et al. 2017, Xiao et al. 2015 and Guenet et al. 2010) in lines 362-373 of the previous manuscript (L.405-415 of the current manuscript). We propose to keep our formulation on these aspects.

Concerning the reviewer's question to specify the size of the ryegrass pieces: they were not homogenously cut, we applied non-refined ryegrass with largest pieces between 10-15 cm long. It is not possible to prove whether or not the added residue was partially incorporated in aggregates, but we do not discard that when EOM was added at 5 g kg$^{-1}$ in the silt loam soil some degree of physical protection might have occurred given the lower relative degradation at this highest input. Confirmation of this remains necessary, as explained in L388-392 of the manuscript.

4. The introduction and motivation of the study is based on very, very old SOC models from 1945. This is clearly not up to date. What about e.g., the AMG model (1999) or the roth-C model? I admit that they are also quite old, but they are at least still used and updated. The same goes for the literature on aggregation processes. Although these are fundamental aspects of soil science and it is therefore fine to cite the fundamental (and old) papers, they are also topics of current science and the knowledge has developed tremendously in recent years. In addition, the authors claim on several occasions that certain aspects have not yet been studied, while neglecting the existing literature on this. E.g., (a) (L.70) The authors claim that no general comparison of the size of organic residues and their decomposition has been done. This is simply not true, as there are several experiments as well as recent modelling studies on this. (b) (L.425) Mapping O2 availability in pores is indeed difficult. However, there are several studies trying to do this by combined approaches of µCT and modelling or micro-sensors. The authors should try to include this literature instead of suggesting that this has not been done. By not including it, one might fear that the authors are not aware about recent developments in research on soil structure, aggregation and C cycling.

We have deleted the sentence and reference to the model by Hénin and Dupuis 1945. We do want to point out that in fact the Roth-C model and DNDC models were already referred in the introduction (L.47-48). We have now included the suggested AMG model (and actually this simple model is also based on the Hénin and Dupuis formalism, and it also follows first-order kinetics (L.47)).

Concerning the comment on the paragraph starting at L.70 of the previous manuscript, namely that the relation between EOM size and its mineralization has already been widely studied, we agree with it. However, as mentioned a few times in this paragraph, here we refer specifically to EOM application dose experiments. In these existing studies on application dose only simple molecules such as glucose were used, or very strongly refined exogenous organic matter, which is not at all representative for field conditions. So that is why we opted to use non-refined plant material as a closer approximation to field conditions and this, to our knowledge, has not been included in experiments investigating the EOM application dose effect on C dynamics.

Regarding L.425, we appreciate this comment. Previously, we did not want to elaborate more on this hypothesis, but following the reviewer's comment, we have now expanded this and added a couple of references as follows in L.485-489: *"As a first step, modelling of soil O₂ transport in 3D soil models based on soil pore models derived from CT volumes could provide further insights in the link between SOC mineralization and pore structure (Schlüter et al., 2022), but such an*

*approach then needs to be validated, e.g. through integration of O₂ microsensors with pore metrics derived from X-Ray µCT, as recently demonstrated by Rohe et al. (2021)."*

Some smaller and more specific comments:

- Soil redox potential: I don't really understand why you expect anoxic conditions in an aerated soil adjusted to a water content of 55% water filled pore space? I understand that this might be the case in fine pores, but you can only rely on bulk soil measurements. Can you explain this in more detail?

We agree that a high Eh measured by larger Pt-electrodes does not necessarily exclude occurrence of local O₂ depletion and demand for alternative electron acceptors. A low bulk soil Eh on the other hand does point at likewise local lower Eh and hints towards O₂ shortage impeding mineralization. There was a clear and consistent difference between the sandy loam and silt loam soil which we deem sufficiently noteworthy, in spite of limitations to using larger Eh electrodes. In any case, we are aware that collected Eh data need to be interpreted with care. We feel that we already sufficiently nuanced our Eh data in the original discussion text, wherein in fact a plea was made to investigate further using O₂ microsensors. We propose to remain with the original formulation.

- L.71: I don't see why the general processes of decomposition should be different in the lab

We explained right after L.71 that in laboratory incubations as the residues are usually finely chopped, a larger residue surface area is in contact with soil mineral surface and more of the added EOM could end up physically protected in small aggregates. Both conditions probably render such lab incubations little representative of field conditions. We already dedicated an entire paragraph to explain these points (L.65-80 of the previous manuscript and L.72-87 in the current version) and do not propose to further expand on this.

- L.110: It is not correct to say that the "initial soil structure is intact" when the soil has been sieved to <10 mm. By this, the original soil structure is completely disrupted. The aggregates might be preserved, but they are only one part of the soil structure.

Indeed, we have reformulated that sentence to focus on preservation of soil aggregation as the original idea (L.132-134).

- The authors describe the study as a macrocosm study. This is not correct in my opinion. I would rather say that it is a microcosm or maybe even a mesocosm study, but never a macrocosm study because it is entirely in the laboratory.

We changed the term "macrocosm" by "mesocosm" throughout the manuscript.

- Pore size distribution: When were the pore sizes measured? At the beginning or end of the experiment?

We measured pore neck size distribution on day 45 of the experiment. We pointed that out now (L.219).

- While I do not question the results of course, I am nevertheless surprised about the low total pore space and amount of coarse pores in the sandy loam. Usually a sandier texture is connected with more coarse pores. Could you please elaborate a little, why the situation could be like this in your soils?

Probably because the silt loam soil led to more macro and microaggregation where at the same time fine pores (within microagreggates) and coarse pores (inter macroaggregates) were formed in the silt loam soil. However, elaborating more on this would be too speculative given that we do not have data on aggregation to support this hypothesis. Moreover, our focus is rather on the effect of application dose and its effect on pore size distribution and C mineralization, i.e. the aspects that have been discussed in our manuscript.

- L.340: You discuss NO3- data. Where can this data be found?

It is presented in figure 7 (L.340), we have also clarified the y-axis title inserting "$NO_3^-N$". We also dedicated a paragraph to it in the methodology section (L.203-208).

- L.390: It is no news that SOC quality and texture influence SOM mineralization and priming. There are studies about that (which especially disentangle these processes)

We agree that there are plenty of studies on the effect of soil texture and SOC mineralization. However, to the best of our knowledge, there are not many studies where EOM application dose has been studied in combination with soil texture. The studies that we have come across concerning application dose and priming of SOC are just mentioned and discussed in the previous paragraph (L.405-413).

- L.388: If you relate the effects on SOC quality, it is even more striking that you don't even mention the crops of the fields and did not test any SOC quality parameters

We understand the comment from the reviewer. We have now included the crop history of the fields (L.125-130); however, logistically it would have been a bit difficult to measure SOC quality when there were many other variables measured already to support our hypotheses.

- L.395: I would delete the whole paragraph. The authors mention themselves that they do not have data about repeated doses and the priming effect was very short-lived. Therefore, I cannot understand how you can recommend a certain soil carbon management under field conditions with repeated doses vs. one time application.

We agree that our reflection on consequences of splitting OM-doses across multiple applications needs to be tested experimentally, as also stated as such in the discussion, which is also why we kept it brief. However, we do think that this is a very relevant point to raise given that it concerns management of OM on the majority of agricultural land. We would thus prefer to keep our current

and more tentative formulation based on the C balance (L.451-453), but leave it up for the editor to decide.

- L.412: You cannot discuss changes in the pore network with a one-time measurement. It is obvious that litter causes a coarser pore space. There is literature about that. You don't even mention if the pore size measurement has been at the beginning or end of the experiment.

We have clarified now that the pore neck size distribution was measured at day 45 of the experiment (L.219). Measuring it in the middle of the incubation appeared to be a good choice given that priming of SOC occurred mainly during the first three weeks of incubation (Fig. B1) and EOM mineralization occurred mainly before 45 days, especially in the silt loam soil (Fig. A1). The reviewer's observation that changes in soil pore network cannot be detected with a single measurement in time is indeed valid. We thus removed any mention of "changes" in soil pore size distribution into relative differences compared to the unamended control soil throughout the manuscript.

- L.435: Although speculative, it is a genuinely very interesting result that the coarse pores and better aeration probably counterbalanced the low redox potential due to O2 consumption. If the paper is resubmitted, it would be nice to elaborate this discussion a bit more.

We complemented the original discussion on this aspect with the sentence *"A more systematic approach combining microscale soil Eh measurements, soil pore network structure and soil respiration, would enable confirmation of this potentially interesting link between Eh and porosity, and their resulting effect on SOC dynamics."* in L.501-503.

**Reviewer #2: Julia Schroeder**

The authors of the manuscript SOIL_2024-107 investigated how the amount of exogenous organic matter (EOM) added to two different textured soils (sandy loam, silt loam) affects its decomposition and priming of SOC using 13C-labelled ryegrass. The authors expected that EOM mineralisation at high dose would be limited in silt loam due to lower $O_2$ availability in microbial hotspots (strong consumption), whereas this limitation would not occur in sandy loam. This would result in an asymptotic increase in EOM mineralisation with dose in silt loam, and a linear increase in sandy loam. However, effects of EOM dose on soil structure may counteract this limitation. The focus of this study lies on potential feedback mechanisms of EOM addition on decomposition by effects on soil structure (pore volume) and thus $O_2$ availability.

The authors find that EOM mineralisation increases with EOM dose, where EOM mineralisation rate is decreasing at high dose in silt loam, but not in sandy loam, which would indicate a limitation of EOM in fine-textured soils as hypothesised. However, they misinterpret their results in the latter case (see comment major concerns 1). SOC priming was found to be higher at higher EOM dose, with larger stimulation in silt loam (see comment major concern 2). They find that EOM addition increases the pore volume in silt loam but not in sandy loam, while Eh (indicator for redox-potential and thus $O_2$ availability) was not altered by EOM dose in none of the soils. They conclude that the effect of EOM dose on macroporosity in silt loam had stimulated SOC priming at high dose. MBC

was increased in both soils with EOM dose, but the authors state that the per unit increase in MBC was higher at lower EOM dose in both soils. Again, this is in my opinion a very brave interpretation of the non-significant results (see comment major concern 3). Taken together, they conclude that the same amount of EOM applied in several smaller doses of EOM would result in less C losses due to mineralisation as compared to the same dose being applied once.

The investigated research question is of broad interest and adds to the current discussions. Understanding how application dose will affect the fate of C entering the soil, has also implications for agricultural recommendations.

As reviewer #1 indicated, the authors published similar work in Biology and Fertility of Soils. The submitted manuscript, however, presents a new laboratory experiment with a new focus and research question, emerging from the results of the previous study. Therefore, it is novel. However, I do agree that the authors should better introduce the novelty of this study in comparison to Mendoza et al. 2022b (e.g. 3 doses instead of 2, microbial biomass determined, larger ryegrass fragments, only <10mm sieved).

Overall, the writing style is clear and easy to follow. But introduction could be further improved to clearly point out the novelty of this study and logically introduce the hypotheses. Adding a figure showing expected relationships between EOM mineralisation and dose for silt loam (asymptotic) and sandy loam (linear) will help to clarify the still somewhat confusing hypotheses. I highly appreciated the detailed M&M section, which allowed to follow the experimental design and indicated in most cases the reasoning behind decisions. However, reasoning for why different statistical approaches were used is lacking and should be added during revision. I have several major concerns regarding the interpretation and thus discussion of the results.

We appreciate the reviewer's comments on the potential impact of the study and addition to SOC dynamics, its novelty as well as the comments that have helped us greatly improve the manuscript.

Concerning the novelty of the study, following the reviewer's comment, we have included a few lines to clarify its novelty compared to Mendoza et al. (2022). The suggested points are now included in the current version of the manuscript (L.88-97) as follows: *"To account for these shortcomings and to obtain more realistic understanding of the application dose effect on EOM and SOC mineralization as it might occur in the field, we aimed to study decomposition of large pieces of $^{13}$C-labelled ryegrass residue in large relatively less disturbed soil cores. Consequently, as compared to Mendoza et al. (2022b) and Shahbaz (2017a), the soil masses used in a newly designed soil incubation experiment were about 23 times and 70 times larger, the soils were only coarsely sieved (<10 mm), the added crop residues were not chopped. Concentration of added ryegrass dose to non-refined ryegrass pieces is expected to yield a stronger local impact on soil structure and the concentration of C mineralization into such relative hotspots might more readily result in local depletion of $O_2$ with potential negative impacts on EOM and SOC mineralization as compared to previous work on this topic."*

We also now included one figure with the hypotheses and expected outcomes in the introduction to make it more understandable for the readers, as suggested by the reviewer.

[Figure]

**Figure 1.** Expected outcomes and hypotheses of the EOM application dose effect on EOM and native SOC mineralization in a sandy loam and a silt loam soil.

Regarding the statistical approach, these comments are specifically covered further below in specific responses.

Major concerns:

- Large parts of the discussion are based on the interpretation of Figure 1, that the proportion of EOM mineralised decreases exponentially with increasing EOM dose. However, this conclusion cannot be made based on the study results. The authors find that EOM mineralisation increases linearly with EOM dose (Fig 1 top). The proportion of EOM C mineralised over EOM dose (shown in Fig 1 bottom) is basically the slope of EOM mineralisation over EOM dose (shown in Fig 1 top). If EOM mineralisation is linear, the slope of this function cannot decrease. Furthermore, while statistical metrics indicate high significance of the regression in Fig 1 top (i.e. the linear relationship between EOM mineralisation and EOM dose seems very solid), the relationship in Fig 1 bottom between the proportion of EOM mineralised and EOM dose is only marginally significant and should be therefore treated even more carefully. Therefore, the results only allow to conclude that EOM mineralisation increases proportionally with EOM dose in both soils, with different proportions of EOM being mineralised between sandy loam and silt loam. Reasoning should be given, why you decided to include only 3 different application doses instead of 4 to 5 equally distributed EOM doses. Including only 3 doses, where two are relatively close and one is far above the two other doses on the x-axis, makes it IMO impossible to find differences between linear and asymptotic fit and thus test your hypothesis of limitation of EOM mineralisation at high dose in silt loam.

We appreciate the comment of the reviewer. However, we disagree that only looking at cumulative EOM mineralization as a function of EOM dose suffices to draw conclusions. This because even though EOM mineralization responds linearly to dose, it is well possible that the mineralized % at a high dose is simply different than the smaller doses. This proved to be the case for the silt loam soil in the respective figure. To illustrate our point that perfect linearity does not always imply a lack of trend in % EOM mineralized, we are presenting the following hypothetical data (with no units for simplicity), and indeed with a linear response of cumulative EOM mineralization to EOM dose, the % mineralized EOM nevertheless decreased.

| | Example 1 | | Example 2 | |
|---|---|---|---|---|
| Dose | Cum. EOM | % min. EOM | Cum. EOM | % min. EOM |
| 2 | 1 | 50.0 | 1.1 | 2.2 |
| 2 | 1 | 50.0 | 0.9 | 1.8 |
| 2 | 1 | 50.0 | 1 | 2.0 |
| 6 | 2 | 33.3 | 2.2 | 6.6 |
| 6 | 2 | 33.3 | 2 | 6.0 |
| 6 | 2 | 33.3 | 1.8 | 5.4 |
| 10 | 3 | 30.0 | 3.1 | 10.3 |
| 10 | 3 | 30.0 | 3.5 | 11.7 |
| 10 | 3 | 30.0 | 3.4 | 11.3 |
| | | | | |
| $R^2$ | 1 | 0.87 | 0.97 | 0.65 |
| P value | <0.001 | <0.001 | <0.001 | <0.01 |

Thus, for Example 1 where increasing the dose from 2 to 6 brings about increased cumulative EOM mineralization (Cum.) while the % mineralized EOM decreased significantly (i.e. 50, 33.3 and 30%). The second example paints a similar picture but then with some more typical variation among replicates.

Based on this, we would prefer to keep the presentation of relative EOM mineralization response to EOM dose (Fig. 2 lower graph) as currently shown in the manuscript, as it is essential for our further interpretation.

Concerning the reasoning behind the selected doses and as suggested by the reviewer, we added a sentence to the following paragraph (L.136-142) as follows: *"These applications were used as representative for rather low, intermediate, and high EOM doses commonly applied in agricultural practices under field conditions, and they correspond to 1.75, 5.25, and 17.5 Mg ha$^{-1}$ assuming a depth of 25 cm and a bulk density of 1.4 g cm$^{-3}$. The intermediate dose of 1.5 g dry matter kg$^{-1}$ soil furthermore closely represents the typical application rate of 2.6 Mg C ha-1 yr$^{-1}$ in German croplands (Riggers et al., 2021). The experiment was limited to only three doses mainly because the large dimensions of the soil mesocosms (6.5-7 kg dry soil per pot) already required considerable amounts of $^{13}$C-labelled plant material.".*

- SOC priming is displayed as absolute increase in SOC mineralisation as compared to the non-amended control. Instead of showing this absolute priming in Fig 2 bottom, which is redundant with Fig 2 top (normalising for y-intercept; this is why statistics are almost the

same), you should display the proportion of SOC priming as relative to the total cumulative CO2-C. Only this would allow to compare how EOM dose affects priming between both soils. The comparison between absolute priming is misleading, since absolute mineralisation differs between both soils, with higher rates in silt loam. In the end, SOC priming might just be proportional to EOM mineralisation. Therefore, the discussion on why SOC priming is stimulated to a larger extent in silt loam as in sandy loam needs revision.

We appreciate and agree with the reviewer's comment, and we are now presenting "relative SOC priming". We checked this now and found that there was no interaction between texture and EOM dose and on relatively expressed SOC priming. We accordingly modified the interpretation in the results section. The conclusion and the discussion were therefore adjusted according to the observation that relative SOC priming response to EOM dose was not different between both soil textures. We also elaborated more on the newly added C balance calculations. These pointed out that adding EOM at a low dose (0.5 g kg$^{-1}$) has an adverse effect on the SOC balance (worse than when no EOM is added at all), while adding EOM at 1.5 and 5 g kg$^{-1}$ yields a more positive balance when expressed per unit of EOM-C added. Hence, the following discussion was added in L.446-453: *"The net C balance was also suggested to differ with EOM dose in both soils, it was in particular lower when EOM was added at low dose then when no EOM was added at all. The slowed relative EOM mineralization in the silt loam soil that was not seen in the sandy loam soil did not result in an improved net C balance as its impact was counteracted by a relatively stronger SOC priming response to EOM dose in the silt loam than in the sandy loam soil. Priming contributed from around 4% up to 40% of to this overall large SOC mineralization going from low to high application doses in both soils (Table C1) and was thus certainly a non-negligible term in the net C balance. The contrast in C balance per unit of EOM dose followed the order: intermediate > high > low doses, we thus might tentatively conclude that adding EOM at low dose is unfavourable from a C balance perspective."*

Also, we deleted L.456-461 of the conclusion in the previous manuscript concerning priming and modify the text accordingly (L.523-542): *"Formation of MBC was independent of EOM dose; thus, we found no evidence suggesting a more economical growth of heterotrophs at higher substrate doses. We expect that with the generally observed lower bulk soil Eh in the silt loam soil, the slowed relative mineralization of EOM at increasing dose could be related to enhanced occurrence of local O₂ limitation surrounding EOM litter, even though its addition in fact also stimulated macroporosity. Revealing causality and identifying the situations where increased O₂ demand due to enhanced microbial activity at higher EOM dose outweighs the potentially improved gaseous transport from increased macroporosity will require experiments targeting changes in soil structure and include local Eh measurements at the microscale. Tentative C balance calculations finally pointed out that when EOM is added at a low dose, i.e. around 0.5 g kg$^{-1}$, it has a negative impact on SOC. When added at 1.5 or 5 g kg$^{-1}$ a positive effect is expected on the C balance. However, these tentative C balances of this upscale pot experiment should now be confirmed in the field where environmental conditions vary."*

- While results are mostly displayed as cumulative results after 90-days incubation, microbial biomass was only measured once after 45 days. The reasoning for this decision is lacking in the M&M. Also, this needs consideration in combined interpretation of EOM dose effects on MBC (45-days) and EOM mineralisation and priming (90-days). Also the information

is lacking in the Fig 3 caption. However, my major concern regarding MBC is the interpretation of Fig 3 bottom. First, the unit is missing. It is unclear from looking at the figure and reading its caption how EOM-mediated MBC increase was calculated. It is not based on 13C-incorporation. I assume it is calculated similar to SOC priming, i.e. normalising for the y-intercept. On the other hand, this would not explain the low values at 5 g dose. Did you divide the delta MBC (to non-amended control) by EOM dose? Second, the relationship is non-significant. Therefore, the conclusion that the increase in MBC per unit EOM added decreases with EOM dose is not valid.

*We now added a motivation of why we measured MBC after 45 days (L.194-195): "We measured MBC after 45 days because at that point in time most of the SOC priming and EOM mineralization had occurred."* This becomes clear from Fig. A1 and Fig. B1.

The unit has now been added to Fig. 3 (now Fig. 4 in the revised version of the manuscript version). To calculate the EOM-mediated MBC increase the difference between amended and control MBC was divided by application dose. We explained this in the manuscript. In agreement with the reviewer's comment, given that we did not find a statistically significant relation between EOM-mediated MBC increase and EOM application dose, we are changing the related discussion, wherein we originally invoked this MBC response to dose to explain the observed trend in EOM mineralization response to dose for the sandy loam soil. We also now changed the caption and presentation of Figure 7 (now Figure 8), where we synthesize the findings of our experiment, by excluding the suggestion that with increasing EOM dose there was a tendency towards smaller MBC increase following EOM application.

General comments:

Statistics

- Indicate reasoning for why different tests were used. Did you use an ANOVA to test for significant effects of dose x lime in your GLM? In lines 212-214 you state that you tested for normal distribution of residuals. However, with GLM residuals do not need to be normally distributed. Revise this section to clarify.

We now clarified that for data at a single time point, general linear models (GLMs) were used, whereas linear mixed models (LMMs) were used for soil Eh measurements taken over time (L.231-234). To test for significant effects of the interaction between dose and texture, we assessed the significance of the interaction coefficient using the "lm" and "summary" functions, which utilize t-tests. Regarding the assessment of normality, we would like to clarify that as stated in L.231 and 241 we used general linear models and linear mixed models (LMMs) where normality of residuals should be checked. We did not use generalized linear mixed models (GLMMs) which can handle non-normal outcomes. Therefore, we prefer to keep the normality check as stated.

Figures

- Display uncertainties, i.e. confidence intervals. Remove redundant figures (see comments major concerns)

Confidence intervals have been displayed and figures have been modified accordingly.

Figure 5/pore neck sizes

- In general, you missed to clearly indicate in the introduction in which pore neck sizes you expected changes to occur, and what would be the implication of such changes. Figure 5 is hard to interpret. It is unclear that EOM (g kg-1) is y-axis label. Different colours are hard to see and I would prefer to have one simple figure on total pore volume changes over having several pore neck sizes. Consider changing axes and aggregating all fractions to display total pore volume differences with error bars. If you are more interested in showing the distribution of pores normalise the total pore volume to better visualise the shift in proportions.

We felt that postulating the promotion of specific pore size classes with EOM application would have been rather poorly funded. Moreover, EOM addition effects on pore size distribution was only one of the mechanisms regarded to potentially explain EOM dose responses of % EOM mineralization and SOC priming. Hence, we now added that we expected promotion of macroporosity (as observed in our previous study Mendoza et al. 2022) with EOM addition and that this could be linked to SOC priming (L.104).

Concerning the comment on Figure 5 (now Figure 6), we have tried the suggested options by the reviewer, and we consider that the current version as a useful way to show the pore neck size distribution. We have now tried to clearly point out where statistical significances are. Moreover, the reviewer's suggestion to present the total pore volumes per dose treatment is already addressed, as the total length of each bar in the current figure represents the total porosity at each dose. This is one of the reasons we also prefer to keep this figure. However, in agreement with the reviewer, we have adjusted the figure and clarified that the y axis represents application dose.

[Figure]

Figure 7

- Needs revision due to major concerns regarding the interpretation of your results.

This figure (now Fig.8) and its caption have now been modified based on the revision on the major concerns about SOC priming and EOM-mediated MBC increase. The EOM-mediated MBC increase part has been deleted.

[Figure]

**Figure 1.** Overview of the mechanisms shaping the relative EOC mineralization (% min. EOC) and SOC priming effects in response to EOM application dose. In the sandy loam soil (left-hand side), increasing EOM dose supplied energy for microbial growth and extracellular enzyme production, which possibly degraded native SOC (i.e. co-metabolism), but did not affect the mineralized EOM fraction. Soil Eh decreased slightly with increasing EOM dose, but C mineralization remained aerobic as expected in the sandy loam soil. Here, soil structure was not much affected by EOM dose. In the silt loam soil (right-hand side), where $O_2$ diffusion is expected to be inherently constrained, increasing EOM dose induced macroporosity which also compensated the large $O_2$ consumption due to larger microbial growth. Here, mainly co-metabolism could explain the positive priming effects with increasing EOM dose in the silt loam soil. The decreasing percentage of decomposed EOM with increasing EOM application dose observed in the silt loam soil might be related to larger EOM protection within aggregates, but further confirmation of this hypothesis is required.

Conclusion:

The study fits the scope of the journal and presents novel results, adding a piece to the puzzle in understanding C dynamics in soil. We learn that EOM mineralisation does increase linearly with its application and that SOC priming seems to linearly correlate with EOM mineralisation. Regarding the major concerns I have with the interpretation of the results, I recommend that the manuscript can only be accepted after a major revision. Given that I expect the manuscript to undergo large changes during its major revision, I will not add line to line comments to this version.

---

## Referee Report (RR1)

Mendoza et al. have addressed all reviewer's comments in their revision of the manuscript SOIL_2024-107. Now, the novelty of this study as compared to Mendoza et al. (2022b) is better introduced in the introduction. The new Figure 1 helps to provide a better overview on the hypotheses. However, I still highly disagree with the interpretation of Figure 2B and the following interpretations.

Relative EOM mineralisation (Figure 2).

I argued previously that the visualisation and interpretation of a non-linear decrease of the proportion of EOM mineralised at higher amounts of EOM added in SiL soils is misleading (Figure 1 in the original manuscript, i.e. Figure 2 in current version). This point is very crucial because this non-linear decline in EOM-C mineralised per EOM-C added is the fundament of further interpretation by the authors. In their reply, the authors have provided two examples to illustrate that the proportion of EOM does not necessarily need to be equal if cumulative EOM increases linearly with dose.

To further look into this problem, I extracted the data from their plots and fitted two linear regression lines. The first allowing for an intercept (*intercept*), the second one forcing the curve through the origin (*origin*). For SiL soils, this gave me the two functions y=279.59x + 50.195 (*intercept*) and y=292.36x (*origin*). Then I projected the cumulative EOM for 20 doses ranging from 0.5 to 10 g to visualise for myself, how the regression would affect the proportion of EOM mineralised per total EOM added. The first fit, i.e. *intercept*, did indeed result in two non-linear curves for SL and SiL, with opposing trends depending on whether the intercept was negative or positive. The second fit, i.e. *origin*, did not. In the *origin* scenario, the calculation of the proportion of EOM mineralised per EOM added (Figure 2B) gives the slope of the fit of EOM mineralised over EOM added (Figure 2A), which is then equal for all doses. As mentioned in my initial comment, Figure 2B is then the derivative of Figure 2A.

[Figure]

[Figure]

[Figure]

[Figure]

I recognize that the authors' argumentation is valid from a mathematical point of view. An exponential fit of the EOM mineralised proportion over EOM dose is possible only if EOM mineralisation allows for an intercept.

I argue that it is not meaningful to assume that there is an intercept. Indeed, it seems much more logical to fit a model through the origin, assuming that the mineralization of EOM is zero, when no EOM is added. As mentioned before, the fit of the non-linear relationship in Figure 2B is weakly significant, and should not be overemphasised.

Furthermore, the question arises whether the observation of a non-linear fit in Figure 2B would really imply some underlying biological mechanisms or just point to uncertainties of the method, e.g. uncertainty of 13C-label at lower doses vs. higher doses? I would rather interpret the decline as caused by the intercept and the intercept being caused by methodological limitations.

The authors' assumed, that EOM mineralisation would slow down at high dose due to O2 depletion at the OM matter surrounding, limiting microbial activity. Assuming this to be the reason for slowed degradation, one would expect that the proportion of EOM mineralised per EOM added would further decline with dose, i.e. stronger limitation at higher O2 depletion and stop of mineralisation. However, my little calculation exercise (i.e. *intercept*) implies that the proportion of EOM mineralised per EOM added will level off at a certain amount, reaching an asymptote at approximately 50% mineralisation. Furthermore, the authors' state that there they did not observe anaerobic conditions in L 380 "but still Eh remained at levels indicative of aerobic conditions".

Long story short, there is indeed little evidence that the proportion of mineralisation is slowed at high EOM dose in SiL and the authors need to revise Figure 2 and following interpretations.

Other points raised during the first review:

Novelty as compared to previous study published in Biology and Fertility of Soils is now better introduced.

Thanks, for the addition of the reasoning behind the selection of dose levels.

Relative SOC priming plot (Figure 3B).          Revise changes in the paragraph added to L.446-453: "The slowed relative EOM mineralization" - see my argumentation above. Also revise L523-542: "…the slowed relative mineralization of EOM at increasing dose could be related to enhanced occurrence of local O2 limitation surrounding EOM litter, even though its addition in fact also stimulated macroporosity. …"

MBC.          I agree with the revisions. Figure 4 becomes now clearer.

Statistics.          Thanks for clarification. I had confounded GLMMs and GLMs.

Figure 1          I highly appreciate the new figure, which illustrates the expected outcomes and hypotheses of the study. I recommend to add a short description of these to the figure caption to provide a quick overview on the hypotheses.

Line-to-line comments

Abstract

L13: Consider to mention your hypotheses with regards to differences to soil texture already in the abstract.

L 15: Delete "economic". Unclear what you mean by that. Do you mean "no increased microbial growth" or "no increased microbial efficiency" or "no changes in microbial growth efficiency"?

L 17-19: I do not agree with the interpretation that the percentage of mineralised EOM decreased with dose and the following hypothesised mechanisms. See detailed comment. Consider revision.

L 20: Delete "textured". "In both soils" is sufficient.

L 21-25: With regards to the high uncertainty of slowed EOM mineralisation in SiL at high dose, I recommend to revise this part. It seems very speculative.

Introduction

L 88-97: I really like this new passage. The novelty of this study is now better introduced.

L 107: Helpful figure! Doesn't the green line show mineralised EOM instead of % mineralised EOM (curve would start high and become lower at high dose in the latter case)? I recommend to add a short explanation on the hypotheses with regards to differences to soil texture to the caption of Figure 1.

Materials and methods

L 161: Do you mean every 1-2 hours?

L 196: CFE-extraction was done in a 1:2 v/w ratio of soil-to-K2SO4. Why didn't you stick to a 1:4 ratio (Joergensen, 1996)? The ratio may affect the kEC. Can you add a reference?

Results

L 254-256: Revise result that relative EOM mineralisation in silt loam soil slowed down at high EOM dose.

L 279: "the extra amount of SOC mineralized vs. the unamended controls relative to the unamended controls" - delete and in caption of Figure 3.

Discussion

The discussion needs a major revision, given that there is no evidence for a slowed EOM mineralisation with EOM dose in the silt loam soil.

L 364: The revised sentence makes no sense anymore. Check.

L 380: "but still Eh remained at levels indicative of aerobic conditions" - this is another argument against the slowed EOM mineralisation with increased EOM dose.

L 385 "In conclusion, we could not identify the cause of these phenomenon, and further research is required to explore the potential mechanisms leading to a relative temporal stabilization of EOM when added at larger doses." - This further suggests, that the authors' may have misinterpreted their results.

L 444: Add a reference for the 1-3% SOC mineralisation in the field.

L 451-453: Unclear whether this conclusion is backed up statistically? There is no statistical analysis mentioned.

L472-474: Only one regression line was fitted for the relative SOC priming over EOM dose in Figure 3 and only one slope is provided in the Results section. Why are different slopes given here? You can not base your discussion on results you did not present and which are furthermore not significant. Please revise the discussion around this point.

Figure 8:        Please revise the trend in % min EOC in silt loam. Unclear what primed SOC revers to (e.g. relative SOC priming, absolute, relative SOC priming per EOM added). Please clarify.

Conclusion

Needs revision with regards to slowed EOM mineralisation with EOM dose in the silt loam soil.

---

## Author Response (AR2)

Dear Authors,

Both reviewers have kindly agreed to review your revised manuscript. They both acknowledge the improvements made and appreciate the effort you have invested. However, one of the reviewers remains critical of how you have used and interpreted the data on the proportion of EOM mineralization, particularly regarding the use of a non-linear model to fit this data. The reviewer presents a compelling argument in their review. In light of this, I have decided to request a major revision of this section of the manuscript, including reconsideration of the model used in the current Figure 2B and corresponding changes to the discussion section. Please also address the other comments provided by the reviewers throughout the manuscript and implement the requested changes. I will reconsider your manuscript after reviewing the revised version.

Looking forward reading a new version of your manuscript,

Kind regards.

We appreciate the editor's feedback for allowing us to revise our manuscript. In this rebuttal we provide a point-by-point response to the reviewers and refer to the line numbers of the document with visible changes. We understand the second reviewer's concern regarding the consideration of the origin in cumulative emissions, as no cumulative EOM emissions are expected when no EOM is added. The reviewer's comments on the lack of response of relative EOM mineralization to EOM dose have been addressed, and we have now applied a linear fit through the origin in the updated figure of relative EOM mineralization. The interpretation has been adjusted throughout the manuscript accordingly. We acknowledge that no evidence could be found for a lowered relative EOM mineralization at high dosage in the studied silt loam soil. Additionally, all other points raised by the reviewer have been addressed, as explained in our detailed responses below.

We hope that the changes made are satisfactory and meet the criteria for publication.

Best regards,

The authors

**Reviewer #1:**

The authors perfomed an extensive revision of the manuscript. The major comments have been adequately addressed, which made the manuscript better in my opinion and therefore I recommend the manuscript for publication.

I acknowledge that the authors have now included and discussed the overall C balance in the soils and in particular the effects on the net C balance.

The novelty of the study is now also better explained and the difference to the previous study is made clear.

Many aspects with respect to the SOM quality and soil structure are still not discussed. However, I understand that the data and experimental design do not really support an elaborate discussion on this.

Recommendations for the management of "real soils" are now made with more caution (comment on previous L.395). I still think that short-term lab incubation studies should not lead to management advice for real life, especially when they did not test at all the effect of repeated doses at all, but the effect of a one-time application of different doses. However, with the cautionary statement presented, I think this is fine.

We appreciate the reviewer's time in re-evaluating the revised manuscript and are grateful for the positive feedback and suggestion to publish our work after final revision.

One technical correction:

The abbreviation MBC should be explained once before it is used in the abstract.

Microbial biomass carbon was now spelled out in full, as this was only mentioned once in the abstract. (L.17)

**Reviewer #2: Julia Schroeder**

Mendoza et al. have addressed all reviewer's comments in their revision of the manuscript SOIL_2024-107. Now, the novelty of this study as compared to Mendoza et al. (2022b) is better introduced in the introduction. The new Figure 1 helps to provide a better overview on the hypotheses. However, I still highly disagree with the interpretation of Figure 2B and the following interpretations.

We are grateful to the reviewer for the considerable amount of time and energy invested in evaluating our manuscript and for the constructive comments, which we believe have helped to improve its quality. We believe that with these revisions, the main concern raised by the reviewer, i.e. no clear evidence for a dosage response on EOM mineralization in the silt loam soil, has now

been adequately considered (please see the addresses to the referee's comments below for further detail).

Relative EOM mineralisation (Figure 2).

I argued previously that the visualisation and interpretation of a non-linear decrease of the proportion of EOM mineralised at higher amounts of EOM added in SiL soils is misleading (Figure 1 in the original manuscript, i.e. Figure 2 in current version). This point is very crucial because this non-linear decline in EOM-C mineralised per EOM-C added is the fundament of further interpretation by the authors. In their reply, the authors have provided two examples to illustrate that the proportion of EOM does not necessarily need to be equal if cumulative EOM increases linearly with dose.

To further look into this problem, I extracted the data from their plots and fitted two linear regression lines. The first allowing for an intercept (*intercept*), the second one forcing the curve through the origin (*origin*). For SiL soils, this gave me the two functions y=279.59x + 50.195 (*intercept*) and y=292.36x (*origin*). Then I projected the cumulative EOM for 20 doses ranging from 0.5 to 10 g to visualise for myself, how the regression would affect the proportion of EOM mineralised per total EOM added. The first fit, i.e. *intercept*, did indeed result in two non-linear curves for SL and SiL, with opposing trends depending on whether the intercept was negative or positive. The second fit, i.e. *origin*, did not. In the *origin* scenario, the calculation of the proportion of EOM mineralised per EOM added (Figure 2B) gives the slope of the fit of EOM mineralised over EOM added (Figure 2A), which is then equal for all doses. As mentioned in my initial comment, Figure 2B is then the derivative of Figure 2A.

[Figure]

I recognize that the authors' argumentation is valid from a mathematical point of view. An exponential fit of the EOM mineralised proportion over EOM dose is possible only if EOM mineralisation allows for an intercept.

I argue that it is not meaningful to assume that there is an intercept. Indeed, it seems much more logical to fit a model through the origin, assuming that the mineralization of EOM is zero, when no EOM is added. As mentioned before, the fit of the non-linear relationship in Figure 2B is weakly significant, and should not be overemphasised.

Furthermore, the question arises whether the observation of a non-linear fit in Figure 2B would really imply some underlying biological mechanisms or just point to uncertainties of the method, e.g. uncertainty of 13C-label at lower doses vs. higher doses? I would rather interpret the decline as caused by the intercept and the intercept being caused by methodological limitations.

The authors' assumed, that EOM mineralisation would slow down at high dose due to O2 depletion at the OM matter surrounding, limiting microbial activity. Assuming this to be the reason for slowed degradation, one would expect that the proportion of EOM mineralised per EOM added would further decline with dose, i.e. stronger limitation at higher O2 depletion and stop of

mineralisation. However, my little calculation exercise (i.e. *intercept*) implies that the proportion of EOM mineralised per EOM added will level off at a certain amount, reaching an asymptote at approximately 50% mineralisation. Furthermore, the authors' state that there they did not observe anaerobic conditions in L 380 "but still Eh remained at levels indicative of aerobic conditions".

Long story short, there is indeed little evidence that the proportion of mineralisation is slowed at high EOM dose in SiL and the authors need to revise Figure 2 and following interpretations.

We agree that a regression through the origin, rather than with an intercept, is more appropriate given the expectation of zero cumulative emissions when no EOM is added. Therefore, we have now fitted a linear model through the origin for cumulative emissions, and the interpretation of the resulting slopes has been updated in both the results section and the figure caption (L.257-263 and L.270-273). Even though there still appears to exist a trend of slowed EOM mineralization at high doses we no longer use this view in our interpretation and discussion. We acknowledge that more evidence (more doses) would have been required to conclude that the proportion of EOM mineralization would indeed decrease at high EOM doses in the silt loam soil. We have accordingly revised our interpretation. In line with this, we have removed the confidence intervals, model equation and determination coefficient from the lower panel of Fig. 2 bottom (L. 267) concerning the proportion of mineralized ryegrass C in silt loam soil and sandy loam soils. In the discussion we now also refrained from invoking a series of potential mechanisms to explain the previously misinterpreted trend in relative EOM mineralization.

Other points raised during the first review:

Novelty as compared to previous study published in Biology and Fertility of Soils is now better introduced.

Thanks, for the addition of the reasoning behind the selection of dose levels.

Relative SOC priming plot (Figure 3B). Revise changes in the paragraph added to L.446-453: "The slowed relative EOM mineralization" - see my argumentation above. Also revise L523-542: "…the slowed relative mineralization of EOM at increasing dose could be related to enhanced occurrence of local O2 limitation surrounding EOM litter, even though its addition in fact also stimulated macroporosity. …"

The mentioned lines 446-453 the previous manuscript version have now been revised (now L.461). We also omitted two sentences referring to the statistical significance of EOM dose on the C balance. A new sentence was added "Adding a low EOM dose was least favourable for the C balance at least in the sand loam soil ($P$ <0.01), while no significant effect of EOM dose on the C balance was observed in silt loam soil." (L458-460).

The second sentence in the conclusion has been revised, – please see our response below in the Conclusion section.

MBC. I agree with the revisions. Figure 4 becomes now clearer.

Statistics. Thanks for clarification. I had confounded GLMMs and GLMs.

Figure 1 I highly appreciate the new figure, which illustrates the expected outcomes and hypotheses of the study. I recommend to add a short description of these to the figure caption to provide a quick overview on the hypotheses.

We thank the reviewer, and we agree to add such further description of the hypotheses - Please see our response below on 2 related remarks on the introduction (L.88-97 and L.107 of the previous manuscript version).

Line-to-line comments

Abstract

L13: Consider to mention your hypotheses with regards to differences to soil texture already in the abstract.

We agree and added the following sentence: *"The percentage of mineralized EOM was expected to increase linearly with EOM dose in sandy loam soil and level off in silt loam soil due to limited $O_2$ supply to maintain aerobic microbial activity."* in L.13-15

L 15: Delete "economic". Unclear what you mean by that. Do you mean "no increased microbial growth" or "no increased microbial efficiency" or "no changes in microbial growth efficiency"?

We can see that this term was unclear, and the sentence has now been adjusted to: *"Likewise, formation of microbial biomass carbon was proportional to the EOM dose, suggesting no reduction in microbial growth efficiency at higher C concentrations"* in L.17-18.

L 17-19: I do not agree with the interpretation that the percentage of mineralised EOM decreased with dose and the following hypothesised mechanisms. See detailed comment. Consider revision.

We acknowledge the referee's points – see our overall response above and in-depth responses to comments raised in the discussion section. We also adjusted the formulation in the abstract and the sentence reads now: *"In the silt loam soil, a decreasing tendency in the percentage of mineralized EOM was apparent but could not be confirmed statistically. We therefore conclude that as in the sandy loam soil the proportion of EOM mineralization was not affected with increasing dose…"* in L.18-21. We have also added a sentence "Consistent with this lack of response in the proportion of EOM mineralization to EOM dose, *soil Eh did not decrease with increasing EOM dose, indicating no $O_2$ limitations.*" in L.23-24.

L 20: Delete "textured". "In both soils" is sufficient.

Done

L 21-25: With regards to the high uncertainty of slowed EOM mineralisation in SiL at high dose, I recommend to revise this part. It seems very speculative.

*We agree that the sentence "At the same time the higher microbial activity might have sufficiently lowered soil Eh close to the large added EOM particles, limiting its relative degradability at high dose, suggesting a potential new mechanism for understanding SOC cycling." was speculative and therefore removed it.*

*We also further slightly rephrased the preceding sentence more conditionally: "The observed stimulation of soil macroporosity at higher EOM doses in the silt loam soil might have contributed to sustaining aerobic conditions required for SOC mineralization." in L.28*

Introduction

L 88-97: I really like this new passage. The novelty of this study is now better introduced.

L 107: Helpful figure! Doesn't the green line show mineralised EOM instead of % mineralised EOM (curve would start high and become lower at high dose in the latter case)? I recommend to add a short explanation on the hypotheses with regards to differences to soil texture to the caption of Figure 1.

*The plotted lines are showing the % mineralized EOM, not cumulative EOM mineralization. We expected the proportion of EOM mineralized to increase with dose in the sandy loam soil and flatten at high doses in the silt loam soil. To clarify further this, we slightly adjusted L.100. "We hypothesized that the mineralized percentage of added EOM (further referred to as relative EOM mineralization) would increase with increasing application dose, due to closer contact of EOM and decomposers." . +*

*We also added a description of the hypotheses to the caption and clarified the expected mechanisms as follows: "**Figure 1.** Expected outcomes and hypotheses regarding the effect of EOM application dose on EOM and native SOC mineralization in sandy loam and silt loam soils. Overall, we expect the proportion of EOM mineralization to increase with higher EOM doses in sandy loam soil due to closer contact between EOM and microbes at higher doses. In silt loam soil, the proportion of EOM mineralization is expected to level off at higher doses, due to a higher chance of limited $O_2$ supply, which may restrict aerobic microbial activity compared to sandy loam soil. Priming of SOC mineralization is expected to increase with higher EOM doses because of enhanced co-metabolism and formation of macroporosity." in L.109-114.*

Materials and methods

L 161: Do you mean every 1-2 hours?

*We meant every 1-2 seconds with the temporal resolution of measurement of $CO_2$ emitted and recorded for about 10 minutes per each soil core. This was already stated in L.166-167.*

L 196: CFE-extraction was done in a 1:2 v/w ratio of soil-to-K2SO4. Why didn't you stick to a 1:4 ratio (Joergensen, 1996)? The ratio may affect the kEC. Can you add a reference?

We have updated a reference for the used method, we now cited Carter and Gregorich 2008 instead of Vance et al., 1987 in L.199. The method proposed by Carter and Gregorich 2008 applies to extraction ratios from (oven dry) 1:2 to 1:5 with no further impact on the extraction efficiency mentioned. Our 1:2 ratio - moist soil:$K_2SO_4$ would in fact result in about a 1:2,5 dry soil : $K_2SO_4$ ratio, i.e. within the proposed range proposed. Tate (1988; Soil Biol. Biochem.1988 20 329335) compared 1:2.5 and 1:5 ratios and found no effect onto extraction efficiency for both statistically significant difference using two extractant-to-soil ratios.

Results

L 254-256: Revise result that relative EOM mineralisation in silt loam soil slowed down at high EOM dose.

This part has now been modified to *"The relative fraction of added EOM mineralized after 90 days was independent of soil texture. However, in the silt loam soil, the relative fraction of mineralized EOM tended to decrease with increasing EOM dose (Fig. 2 bottom). Given the limited number of EOM doses included and close linear response of the cumulative EOM mineralization to EOM dose (with an intercept of zero due to the absence of cumulative EOM mineralization when no EOM is added), this trend should be interpreted with caution."* in L.256-262.

L 279: "the extra amount of SOC mineralized vs. the unamended controls relative to the unamended controls" - delete and in caption of Figure 3.

The sentence is now deleted as suggested. The remaining text is slightly adjusted accordingly, and now reads:*"The lower graph compares the relative priming of native SOC mineralization in both soil textures."* (L.284-285).

Discussion

The discussion needs a major revision, given that there is no evidence for a slowed EOM mineralisation with EOM dose in the silt loam soil.

As explained in our overall response above, we agree and revised the discussion in the following points:

1° We acknowledge that there is no proof to assume a dosage response in the reformulated text.

2° We reorganized the second paragraph of the discussion in which we previously discussed several potential explanations for an EOM mineralization dosage response in silt loam soil. The first part of that discussion on soil N availability has been removed (L.371-375), and now it is just briefly mentioned under section 4.2 to demark the differences in methodological approach between our study and other published work on this topic (L.422-423).

3° We now only concentrate on our actual observation of a lacking response of Eh to EOM dose and refrain now from making any further reference to other mechanisms not under investigation like occlusion of the added EOM into soil aggregates (L.387-394)

Specifically, the following text was changed

*"In the sandy loam soil and silt loam soils, EOM-derived C mineralization was overall independent of its application dose. A decreasing tendency did emerge with increasing EOM dose in the silt loam soil (Fig. 2), did not provide sufficient evidence to support our hypothesis that the relative EOM mineralization would increase with increasing EOM dose"* in L.347-350.

*"The unresponsiveness of relative EOM mineralization to EOM application dose in the sandy loam soil and silt loam soils was consistent with…"* in L.357.

*"Drawing conclusions for EOM management in the field based on this 90-day lab incubation experiment at 20°C is to be made with care. Nevertheless, the ordination of relative EOM mineralization patterns remained consistent among the established dose treatments over time and are projected to remain likewise for at least some time (Fig. A1). For instance, mineralization over 137 days at the established 20°C in the lab experiment equates to about one year in the field in Belgium (9.7°C on average) (De Neve et al., 1996). Thus, our results suggest no, or at most, a limited negative effect of adding EOM at increasing doses on its annual mineralization, a traditionally used metric in C-balance calculations (the so-termed humification coefficient). However, empirical evidence from field experiments is now needed to confirm these findings."* in L.395-403.

L 364: The revised sentence makes no sense anymore. Check.

The EOM-mediated MBC increase was NOT proportional to EOM dose, we have corrected that: *"In our experiment, the EOM-mediated MBC increase was not proportional to EOM dose in both soil textures (Fig. 4; bottom), suggesting that further growth of MBC was energetically equal for the different included substrate concentrations."* in L.364.

L 380: "but still Eh remained at levels indicative of aerobic conditions" - this is another argument against the slowed EOM mineralisation with increased EOM dose. +  L 385 "In conclusion, we could not identify the cause of these phenomenon, and further research is required to explore the potential mechanisms leading to a relative temporal stabilization of EOM when added at larger doses." - This further suggests, that the authors' may have misinterpreted their results.

The discussion has been revised and several sentences were adjusted. L.385-387 now reads: *"Hence, from the Eh readings, we observed no indication that $O_2$ limitations would have restricted the relative EOM mineralization in the silt loam soil at higher EOM doses as we initially hypothesized."*

L 444: Add a reference for the 1-3% SOC mineralisation in the field.

We now refer to Vleeshouwers and Verhagen (2002) wherein annual SOC mineralization for several estimates for Western European countries are listed. Based on that study we also modified the previously given 1-3% into 2-3% in L.455 (Vleeshouwers & Verhagen, 2002; Global Change Biology 8, 519-530).

L 451-453: Unclear whether this conclusion is backed up statistically? There is no statistical analysis mentioned.

It is, and we now reformulated the sentence and added a p value: *"Adding a low EOM dose had the least favourable effect on the C balance at least in the sandy loam soil (P <0.01), while no significant effect of EOM dose on the C balance was observed in the silt loam soil"* in L.458-460.

L472-474: Only one regression line was fitted for the relative SOC priming over EOM dose in Figure 3 and only one slope is provided in the Results section. Why are different slopes given here? You can not base your discussion on results you did not present and which are furthermore not significant. Please revise the discussion around this point.

We acknowledge that the slopes for each texture should not be reported here, as they were not significantly different. Therefore, we have removed the corresponding sentence (L.480-482) and revised the text accordingly as follows: *"We furthermore found positive linear relationships (via linear regressions) between the silt loam soil volume fraction of pore neck size classes 60–100 and >300 μm and relative SOC priming ($R^2$ = 0.34 and 0.36; and, P = 0.09 and 0.08, respectively), and a negative relation with the 3-9 μm class that also depended on EOM dose. We therefore hypothesize that the development of macroporosity might have contributed to the promotion of relative SOC priming in the silt loam soil. In contrast, no such relationships existed for the sandy loam soil and the observed increase in relative SOC priming with EOM dose must have been mediated by other mechanisms."* in L.483-489.

Figure 8: Please revise the trend in % min EOC in silt loam. Unclear what primed SOC revers to (e.g. relative SOC priming, absolute, relative SOC priming per EOM added). Please clarify.

The picture depicts relative SOC priming and that has now been clarified in the caption (in L.519). Moreover, the % EOM mineralized has been adjusted with the new interpretation of no EOM dose effect on relative EOM mineralization (L.525-526).

Conclusion

Needs revision with regards to slowed EOM mineralisation with EOM dose in the silt loam soil.

The conclusion is now revised, and the main adjusted parts read:

*"Overall, our results showed no response of relative EOM mineralization to EOM dose in heavy- nor in light-textured soil, in line with a null response of MBC formation to EOM dose. A large range of doses and soil textures including clay and sandy soils would help clarify if such a dosage independency is consistently observed. Our experiment revealed a lower bulk soil Eh in the silt loam soil than in the sandy loam soil, as expected. However, since Eh remained within the aerobic*

*range even at a high EOM dose, it suggests that $O_2$ supply was sufficient to sustain the proportionally higher absolute EOM mineralization. We hypothesize that the enhanced macroporosity at the established higher EOM doses may have improved soil aeration, preventing the onset of $O_2$ limitations."* in L.531-538.

*Tentative C balance calculations finally indicated that adding EOM at a low dose (around 0.5 g kg-1), had the least favourable effect on SOC at least for the sandy loam soil and no effect in the silt loam soil."* in L.545-547.

The following sentences were removed:

*"The formation of MBC was independent of EOM dose; thus, we found no evidence suggesting a more economical growth of heterotrophs at higher substrate doses."* in L.538-539.

*"We expect that with the generally observed lower bulk soil Eh in the silt loam soil, the slowed relative mineralization of EOM at increasing dose could be related to enhanced occurrence of local $O_2$ limitation surrounding EOM litter, even though its addition in fact also stimulated macroporosity."* in L.539-542.

---

## Author Response (AR3)

Dear Authors

I have reviewed your responses and modifications, and I believe you have done an excellent job addressing the reviewers' comments. The manuscript has significantly improved as a result. I am pleased to accept your submission, pending a minor clarification regarding Figure 1.

For Figure 1, please explain the meaning of "MBC" in the caption. Additionally, the figure appears to show that the soil pore structures in both the sandy loam and silt loam remain similar across all doses. It would be helpful if you could clarify that the figure mainly highlights changes induced by the doses, without considering the initial differences between the two soil types.

Thank you for all your efforts throughout the review process. I look forward to seeing your manuscript published.

Kind regards

We thank the editor for her positive feedback and for her thorough review of our manuscript. We appreciate that our manuscript is close to acceptance for publication.

Regarding Figure 1, we have made the requested clarifications. Specifically:

- We have clarified the MBC meaning in our caption as *"Priming of SOC mineralization is expected to increase with higher EOM doses, although no specific hypotheses regarding soil texture differences are proposed. This increase is attributed to occur because of enhanced co-metabolism, where higher microbial activity -reflected by increased MBC- would promote SOC mineralization, and formation of macroporosity."* in L. 110-113. Accordingly, we have also updated Fig.1 and positioned the lines representing SOC priming at the same level in both soil textures in L.104.
- *We have also clarified that: "The figure illustrates an increase of microbial biomass carbon (MBC) and macroporosity with increasing EOM dose, without making assumptions between soil textures."* in L.106-107.
- Additionally, we have updated all instances of "Eh" in the text and in Fig. 5 (L.310) to "$E_H$" with the correct subscript formatting for "H".

We hope that the changes made are again satisfactory and meet the criteria for publication.

Best regards.

The authors

---

## Author Response (AR4)

Dear authors,

Thank you for altering your Figure 1. I am pleased to accept the publication of your manuscript.

Kind regards

Comments to the author:

On behalf of the editorial team of I am pleased to inform you that your manuscript, titled, has been accepted for publication. We sincerely appreciate the effort and dedication you have demonstrated.

Your article has undergone a thorough review process, and I would like to commend you on addressing the comments and suggestions provided by the reviewers and the topical editor, your careful revisions have significantly enhanced the quality and clarity of the manuscript.

Regards

Dear editors

We appreciate and thank you for your time and effort throughout the manuscript revision process. We are grateful for the acceptance of our manuscript for publication in *SOIL*.

The authors